# Structure of LRRK1 and mechanisms of autoinhibition and activation

Janice M. Reimer[1,2], Andrea M. Dickey ⓘ [1,2,8], Yu Xuan Lin[1,2,8], Robert G. Abrisch[1,2,8], Sebastian Mathea[2,3,4], Deep Chatterjee[2,3,4], Elizabeth J. Fay[5], Stefan Knapp ⓘ [2,3,4], Matthew D. Daugherty ⓘ [5], Samara L. Reck-Peterson ⓘ [1,2,6,7] ✉ & Andres E. Leschziner ⓘ [1,2,5] ✉

Leucine Rich Repeat Kinase 1 and 2 (LRRK1 and LRRK2) are homologs in the ROCO family of proteins in humans. Despite their shared domain architecture and involvement in intracellular trafficking, their disease associations are strikingly different: LRRK2 is involved in familial Parkinson's disease while LRRK1 is linked to bone diseases. Furthermore, Parkinson's disease-linked mutations in LRRK2 are typically autosomal dominant gain-of-function while those in LRRK1 are autosomal recessive loss-of-function. Here, to understand these differences, we solved cryo-EM structures of LRRK1 in its monomeric and dimeric forms. Both differ from the corresponding LRRK2 structures. Unlike LRRK2, which is sterically autoinhibited as a monomer, LRRK1 is sterically autoinhibited in a dimer-dependent manner. LRRK1 has an additional level of autoinhibition that prevents activation of the kinase and is absent in LRRK2. Finally, we place the structural signatures of LRRK1 and LRRK2 in the context of the evolution of the LRRK family of proteins.

ROCO proteins, discovered 20 years ago[1], are distinguished by a Ras-like GTPase embedded in a larger polypeptide, in contrast to G proteins that function independently. This architecture led to the naming of these Ras-like domains as Ras-Of-Complex, or ROC. ROCO proteins are present in bacteria, archaea, plants and metazoans[2]. All known ROCO proteins have an architectural domain immediately following the GTPase, termed a C-terminal Of Roc, or COR. This ROC–COR architecture is what gives rise to the family name, ROCO.

ROCO proteins in humans, of which there are four, gained prominence when mutations in one of its members, Leucine Rich Repeat Kinase 2 (LRRK2), were shown to be a common cause of familial Parkinson's disease (PD)[3–5]. LRRK2 belongs to a class of ROCO proteins that, in addition to its ROC GTPase, contain a kinase domain immediately following the COR domain. Only one other ROCO protein in humans belongs to the same class: Leucine Rich Repeat Kinase 1 (LRRK1), LRRK2's closest homolog. LRRK1 and LRRK2 have a similar domain organization: an N-terminal half containing Ankyrin (ANK) and Leucine Rich Repeats (LRR), and a catalytic C-terminal half containing the ROC–COR ROCO signature, followed by the kinase and a WD40 domain (Fig. 1a). The only difference between them is the presence of an N-terminal Armadillo repeat domain in LRRK2, which LRRK1 lacks. The similarities between LRRK1 and LRRK2 extend to their cell biological functions. Both proteins are involved in intracellular trafficking; they phosphorylate Rab GTPases that mark vesicular cargo transported along the microtubule cytoskeleton by the molecular motors dynein and kinesin[6–8]. However, LRRK1 and LRRK2 phosphorylate non-overlapping sets of Rabs[6–8]. For example, LRRK1 phosphorylates Rab7a, which is involved in the late endocytic pathway,

[1]Department of Cellular and Molecular Medicine, University of California San Diego, La Jolla, CA, USA. [2]Aligning Science Across Parkinson's (ASAP) Collaborative Research Network, Chevy Chase, MD, USA. [3]Institute of Pharmaceutical Chemistry, Goethe-Universität, Frankfurt, Germany. [4]Structural Genomics Consortium, Buchmann Institute for Life Sciences, Goethe-Universität, Frankfurt, Germany. [5]Department of Molecular Biology, School of Biological Sciences, University of California San Diego, La Jolla, CA, USA. [6]Department of Cell and Developmental Biology, School of Biological Sciences, University of California San Diego, La Jolla, CA, USA. [7]Howard Hughes Medical Institute, Chevy Chase, MD, USA. [8]These authors contributed equally: Andrea M. Dickey, Yu Xuan Lin, Robert G. Abrisch. ✉e-mail: sreckpeterson@health.ucsd.edu; aleschziner@health.ucsd.edu

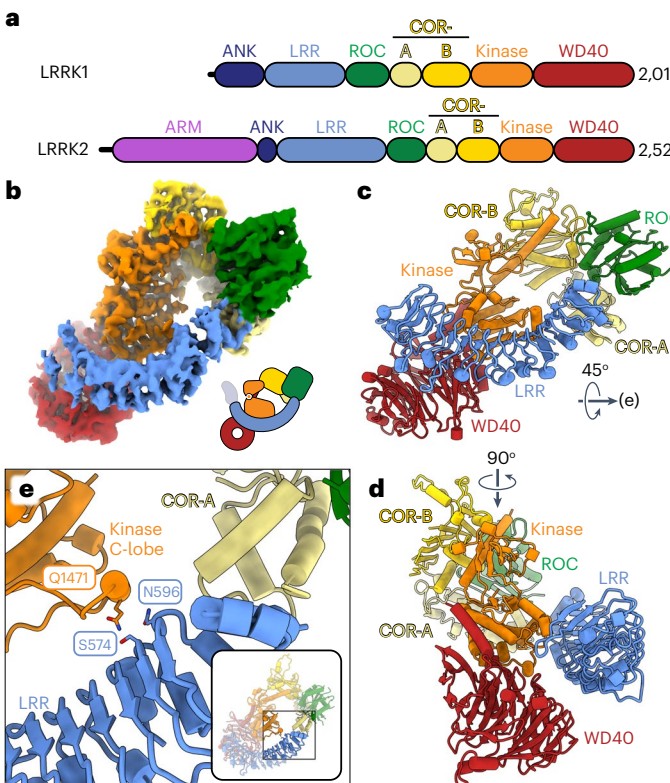

**Fig. 1 | Cryo-EM structure of monomeric LRRK1. a**, Schematic domain organization of LRRK1 and LRRK2. The coloring scheme shown here is used in all figures. **b–d**, Cryo-EM map (**b**) and two views of the model (**c,d**) of monomeric LRRK1. Note: the monomer model shown here is the one built after symmetry expansion of the LRRK1 dimer, as that map showed better density for the region highlighted in **e**. The cartoon in **b** indicates that, while present in our construct, the ANK domain is disordered in the cryo-EM map. **e**, Close-up of the contact between the LRR repeat and the kinase's C-lobe. The inset highlights the region of the model shown in the main panel.

while LRRK2 phosphorylates Rab8 and Rab10, which are involved in *trans*-Golgi transport[9].

Despite the molecular and cell biological similarities between LRRK1 and LRRK2, the proteins are strikingly different in their disease association. All the most common PD-linked mutations in LRRK2 are autosomal dominant gain-of-function mutations that activate its kinase[10]. LRRK2 has also been linked to Crohn's disease and leprosy[11,12]. In contrast, LRRK1 is not involved in PD and is instead linked to two rare bone diseases: osteopetrosis and osteosclerotic metaphyseal dysplasia[13]. In further contrast with LRRK2, disease-linked mutations in LRRK1 are autosomal recessive, loss-of-function mutations[13]. How two proteins with such similar domain architecture and related cellular functions are so different in their involvement in pathology remains a mystery.

Our structural and mechanistic understanding of LRRK2 has expanded substantially over the past few years, with several structures of the protein now available[14–17], along with insights into LRRK2's cellular localization[18], substrates[6–8] and regulation[10]. This level of knowledge is missing for LRRK1, for which only low-resolution structures have been reported for the full-length protein[19]. Understanding the differences and similarities between LRRK1 and LRRK2 would shed light into their unique cellular functions and how those lead to such different involvements in disease. Furthermore, determining what properties are LRRK1 specific would help us better define those that are unique to LRRK2, and thus likely to be involved in the etiology of LRRK2-associated PD.

In this Article, we report cryo-EM structures of full-length LRRK1 in its monomeric and dimeric forms, revealing important differences between the two proteins. LRRK1 and LRRK2 use different mechanisms of autoinhibition, with LRRK1 having a second level of autoinhibition absent in LRRK2. We also perform an evolutionary analysis of LRRK proteins to determine when characteristic structural features of LRRK1 and LRRK2 arose during metazoan evolution.

## Results

### Cryo-EM structure of monomeric LRRK1

We purified LRRK1(Δ1–19), where the N-terminal 19 residues, predicted to be disordered, were deleted, and imaged it in the presence of Rab7a, ATP and GTP. Most particles classified as monomers, with a small subset forming dimers (Extended Data Fig. 1). The monomer particles yielded a 3.6-Å structure of LRRK1(Δ1–19) (Fig. 1b–d, Table 1 and Extended Data Fig. 1). Although the catalytic C-terminal half of LRRK1, which contains the ROC, COR, kinase and WD40 domains ('RCKW'), adopts a J-shaped architecture similar to that of LRRK2 (ref. 17), the location of the N-terminal LRR domain differs between LRRK1 and LRRK2 in a functionally important way. The LRR of LRRK2 physically blocks access to the kinase's active site (Fig. 2a), in what appears to be an autoinhibited conformation[16]. In contrast, the LRR of LRRK1 is shifted towards its WD40 domain, leaving its kinase's active site exposed (Figs. 1b and 2a). The only interaction made by the LRR domain with the rest of LRRK1(Δ1–19) is a contact with the kinase's C-lobe (Fig. 1e). Interestingly, the residue in the kinase involved in this contact corresponds to N2081 in LRRK2, where a mutation linked to Crohn's disease has been identified[11]. The kinase domain of LRRK1 is in the open, or inactive, conformation with its DYG motif (a tripeptide involved in ATP binding) 'out'. Even though the overall conformation of LRRK1(Δ1–19) would not prevent a Rab substrate from being engaged, we did not see any density for Rab7a in our LRRK1 map. This could be explained in part by the presence of an autoinhibitory loop extending from COR-B into the kinase active site, which we discuss below. The ANK domain was not visible in our map.

An unusual feature of LRRK1 is the length of its kinase's αC helix: it is approximately four turns longer than is typical in kinases (Fig. 2b). The extra residues pack against the COR-B domain through an extensive hydrophobic interaction that is unique to LRRK1 (Extended Data Fig. 2a,b). Interestingly, the RCKW moiety of full-length LRRK1 fits well within a map of LRRK1RCKW (Extended Data Fig. 2c), indicating that the presence of the N-terminal repeats does not alter the conformation of the catalytic half of the protein. This contrasts with LRRK2, where the position of the ROC–COR domains differs significantly between the full-length and LRRK2RCKW structures (Extended Data Fig. 2d). It is possible that the more extensive interface between LRRK1's αC helix and COR-B domain rigidifies the RCKW portion of LRRK1. Additionally, the longer helix is involved in one of LRRK1's dimer interfaces, discussed below, which would not be possible with a helix of standard length.

The penultimate propeller blade of LRRK1's WD40 domain has several distinctive features (Fig. 2c). It contains a ~112 residue disordered loop, unique to LRRK1. The third and fourth strands of this blade are unusually long and would clash with where LRRK2's hinge helix interacts with the WD40 domain (Fig. 2c); LRRK1 lacks a hinge helix or an equivalent structural element that can interact with the WD40 domain. LRRK2's hinge helix is inserted at the start of the LRR domain and precedes a ~100 residue disordered loop that contains key phosphorylation sites for binding members of the 14-3-3 family of proteins[20], which are involved in regulating signaling in eukaryotic cells. The analogous, much shorter sequence in LRRK1 (residues 244–255) is disordered in our structure.

The C-terminal helix is a structural feature shared between the two LRRK proteins. In LRRK1, the last six residues of the protein are disordered, resulting in a helix that is shorter than that of LRRK2 (Fig. 2d). It was postulated that for LRRK2 the C-terminal helix, kinase

**Table 1 | Cryo-EM data collection, refinement and validation statistics**

| Part 1 | LRRK1(Δ1–19) | LRRK1(Δ1–19) | LRRK1(Δ1–19) |
|---|---|---|---|
| | | Local refinement around KW | Local refinement around RCKW |
| | (EMD-27813) (PDB 8EO4) | (EMD-27816) | (EMD-27814) |
| **Data collection and processing** | | | |
| Magnification | 36,000 | 36,000 | 36,000 |
| Voltage (kV) | 200 | 200 | 200 |
| Electron exposure (e⁻ Å⁻²) | 51 | 51 | 51 |
| Defocus range (μm) | 1.3–2.1 | 1.3–2.1 | 1.3–2.1 |
| Pixel size (Å) | 1.16 | 1.16 | 1.16 |
| Symmetry imposed | C1 | C1 | C1 |
| Initial particle images (no.) | 724,795 | 724,795 | 724,795 |
| Final particle images (no.) | 69,361 | 69,361 | 69,361 |
| Map resolution (Å) | 3.7 | 3.4 | 3.6 |
| FSC threshold | 0.143 | 0.143 | 0.143 |
| Map resolution range (Å) | 3.5–6.0 | 3.2–6.0 | 3.3–6.0 |
| **Refinement** | | | |
| Initial model used (PDB code) | AlphaFold | | |
| | Q38SD2 | | |
| Model resolution (Å) | 4.0 | | |
| FSC threshold | 0.5 | | |
| Map sharpening B factor (Å²) | 89.9 | | |
| Model composition | | | |
| Non-hydrogen atoms | 10,663 | | |
| Protein residues | 1,343 | | |
| Ligands | 1 | | |
| B factors (Å²) | | | |
| Protein | 88 | | |
| Ligand | 80 | | |
| r.m.s. deviations | | | |
| Bond lengths (Å) | 0.004 | | |
| Bond angles (°) | 0.785 | | |
| **Validation** | | | |
| MolProbity score | 1.33 | | |
| Clashscore | 3.97 | | |
| Poor rotamers (%) | 0.08 | | |
| Ramachandran plot | | | |
| Favored (%) | 97.17 | | |
| Allowed (%) | 2.83 | | |
| Disallowed (%) | 0.00 | | |

**Table 1 (continued) | Cryo-EM data collection, refinement and validation statistics**

| Part 2 | LRRK1(Δ1–19) | LRRK1(FL) | LRRK1(FL) |
|---|---|---|---|
| | Local refinement around RCK | | Symmetry expansion |
| | (EMD-27815) | (EMD-27817) (PDB 8EO5) | (EMD-27818) (PDB 8EO6) |
| **Data collection and processing** | | | |
| Magnification | 36,000 | 36,000 | 36,000 |
| Voltage (kV) | 200 | 200 | 200 |
| Electron exposure (e⁻ Å⁻²) | 51 | 55 | 55 |
| Defocus range (μm) | 1.3–2.1 | 1.3–2.1 | 1.3–2.1 |
| Pixel size (Å) | 1.16 | 1.16 | 1.16 |
| Symmetry imposed | C1 | C2 | C1 |
| Initial particle images (no.) | 724,795 | 396,404 | 396,404 |
| Final particle images (no.) | 69,361 | 58,913 | 117,826 |
| Map resolution (Å) | 3.7 | 4.6 | 4.3 |
| FSC threshold | 0.143 | 0.143 | 0.143 |
| Map resolution range (Å) | 3.5–6.0 | 4.5–6.5 | 4.0–6.5 |
| **Refinement** | | | |
| Initial model used (PDB code) | | 8×10⁶ | 8×10⁴, AlphaFold Q38SD2 |
| Model resolution (Å) | | 6.1 | 4.5 |
| FSC threshold | | 0.5 | 0.5 |
| Map sharpening B factor (Å²) | | 230 | 131 |
| Model composition | | | |
| Non-hydrogen atoms | | 27,732 | 13,871 |
| Protein residues | | 3,498 | 1,749 |
| Ligands | | 2 | 1 |
| B factors (Å²) | | | |
| Protein | | 229 | 137 |
| Ligand | | 185 | 106 |
| r.m.s. deviations | | | |
| Bond lengths (Å) | | 0.004 | 0.004 |
| Bond angles (°) | | 0.73 | 1.079 |
| **Validation** | | | |
| MolProbity score | | 1.41 | 1.35 |
| Clashscore | | 3.74 | 3.49 |
| Poor rotamers (%) | | 0 | 0 |
| Ramachandran plot | | | |
| Favored (%) | | 96.36 | 96.65 |
| Allowed (%) | | 3.64 | 3.35 |
| Disallowed (%) | | 0 | 0 |

N-lobe and COR-B form a regulatory hub where phosphorylation of residue T2524 in the C-terminal helix could regulate kinase activity[14]. There are no known phosphorylation sites on LRRK1's C-terminal helix, and it does not extend far enough to contact the kinase N-lobe or COR-B. Since AlphaFold's[21] (https://alphafold.ebi.ac.uk/) predicted LRRK1 structure has a fully folded C-terminal helix, we wanted to

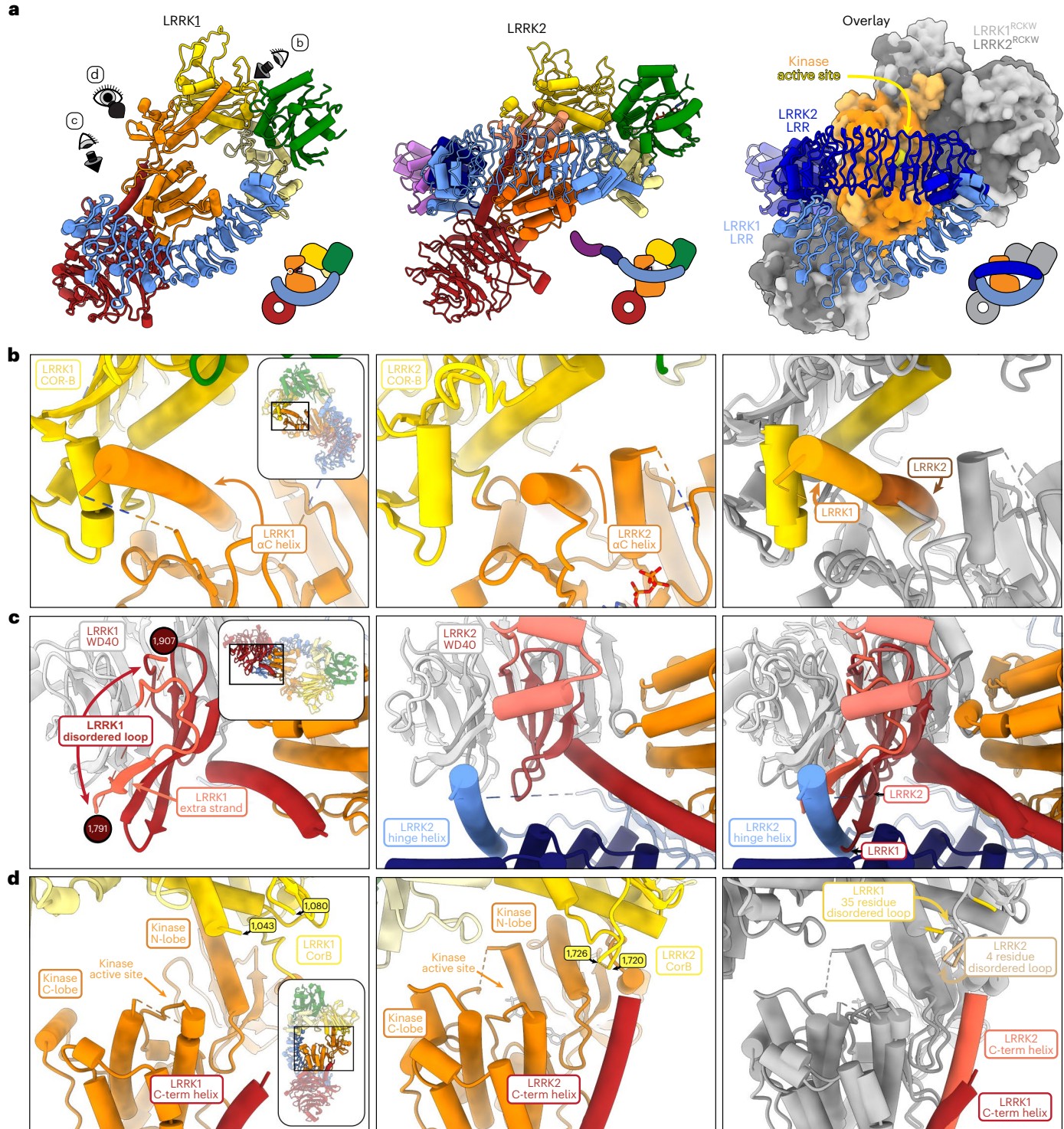

**Fig. 2 | Comparison of monomeric LRRK1 and LRRK2. a**, The LRR domain in LRRK1 does not sterically block access to the kinase's active site as it does in LRRK2. Models and cartoons are shown for LRRK1 (left), LRRK2 (center) and an overlay of the two structures (right). In the overlay, the 'RCKW' portion of LRRK1 and LRRK2 is shown as a surface representation, with the kinase in orange and the remaining domains in gray (light gray for LRRK1, dark gray for LRRK2). The directions of close-ups shown in **b**–**d** are indicated on the left-side panel. **b**–**d**, These panels show comparisons between LRRK1 (left) and LRRK2 (center) focused on features that are different between the two structures, and a superposition to highlight those differences (right). The insets highlight the region of the structure shown in the main panel. **b**, The αC helix in LRRK1's kinase domain is several turns longer than its counterpart in LRRK2. **c**, LRRK1's WD40 domain has features that would clash with an element analogous to LRRK2's latch helix. **d**, LRRK1's C-terminal helix is shorter than LRRK2's.

understand whether this is a result of AlphaFold's LRRK1 being modeled in an active state or a more fundamental feature of unknown function. We deleted the last six residues of LRRK1 [LRRK1(Δ2010−2015)]

and measured phosphorylation of Rab7a, a LRRK1 substrate[8], in cells. We did not observe a significant difference in Rab7a phosphorylation between full-length LRRK1 and LRRK1(Δ2010−2015) (Extended Data

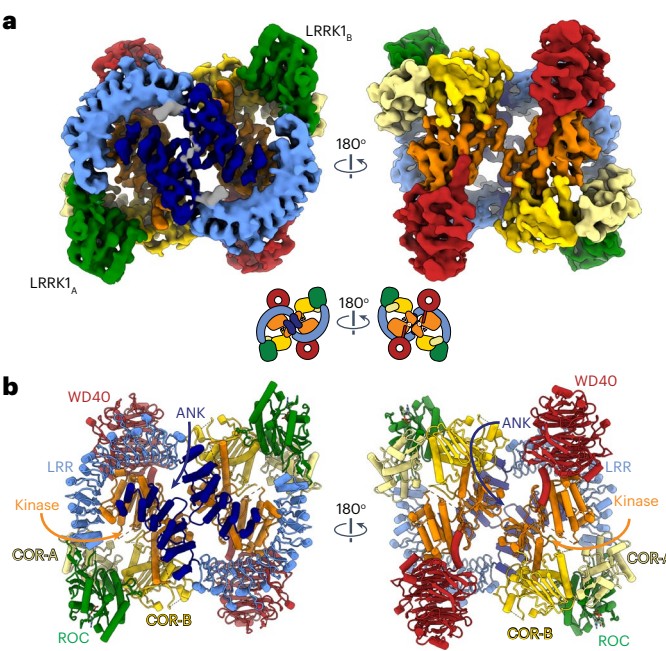

**Fig. 3 | Cryo-EM structure of dimeric LRRK1. a,b,** Cryo-EM map (**a**) and model (**b**) of dimeric LRRK1. Domains are indicated for the LRRK1a monomer in **b**.

Fig. 3), suggesting that the end of the C-terminal helix does not play a major role in LRRK1 regulation. We note that we could not detect LRRK1 in 293T cells in the absence of overexpression (Extended Data Fig. 4a,b), and that no peptides for either LRRK1 or LRRK2 were found in the 293T cell proteome in a previous study[22]. Similarly, we could not detect statistically significant differences in the expression levels of any of the LRRK1 constructs tested throughout this work (Extended Data Fig. 4c–g).

## Cryo-EM structure of dimeric LRRK1

We imaged full-length LRRK1 in the presence of guanosine diphosphate (GDP). Surprisingly, this construct yielded almost exclusively dimers (Extended Data Fig. 5), of which we obtained a 4.6-Å structure (Fig. 3, Table 1, Extended Data Fig. 5 and Supplementary Video 1). Each molecule in the dimer has the same conformation we observed in the monomer; however, the ANK domain is fully resolved in the dimer structure.

Surprisingly, despite the similarities in their domain organization and the structures of the monomers, the LRRK1 dimer bears no resemblance to that of LRRK2 (Fig. 4a,b). While LRRK2 forms a parallel dimer mediated by a single homotypic interaction involving its COR-B domain (Fig. 4b), LRRK1 forms an antiparallel dimer mediated by several homo- and heterotypic interactions (Fig. 4a). The ANK–LRR domains of each LRRK1 wrap around those of the other, making symmetrical contacts between the ANK:ANK and LRR:ANK domains (Fig. 4c,d). While the resolution is too low to determine specific interactions at the ANK:LRR interface, the surfaces involved are electrostatically complementary (Fig. 4d). A major interaction interface is formed by the kinase C-lobe and LRR domains of one monomer and the opposite molecule's ANK domain (Fig. 4e). In contrast to LRRK2, whose dimer is mediated entirely by the COR-B domain in its RCKW moiety, the LRRK1 dimer shows minimal direct interactions between RCKWs; the only contact seen in our structure is a homotypic interaction involving residues 1,263–1,265 in the kinase N-lobe (Fig. 4f).

The recent structures of dimeric LRRK1 (this work) and LRRK2 (ref. 16) all differ significantly from lower-resolution structures reported earlier[19]. It remains to be seen whether those earlier structures represent different dimeric forms of LRRK1 and LRRK2.

## LRRK1 is sterically autoinhibited in trans in the dimer

The main contact between the two LRRK1 monomers, and the most striking feature of the dimer, involves the ANK domains; each ANK domain contacts the opposite molecule's kinase on both its N- and C-lobes (Fig. 4h–k), effectively blocking access to the kinase. Our structure suggests that the LRRK1 dimer uses the ANK domain to accomplish, in *trans*, the steric autoinhibition achieved by the LRR domain in the LRRK2 monomer.

## LRRK1's N-terminus stabilizes the dimer

The LRRK1 (full-length) and LRRK1(Δ1–19) constructs resulted almost exclusively in dimers and monomers, respectively, despite differing only in the presence or absence of 19 residues at the N-terminus. This was particularly surprising as AlphaFold predicts the first ~48 residues of LRRK1 to be disordered (Fig. 5a), and because the first residue we were able to model in the LRRK1 dimer was R51 (Fig. 5b). However, we noted two areas of density in our cryo-EM map, near the junction of the ANK–LRR domains and on top of the ANK:ANK interaction, that were not accounted for by our or the AlphaFold models (Fig. 5c and Supplementary Video 1). We hypothesized that these densities could be accounted for by the extreme N-terminus of LRRK1 stabilizing the autoinhibited dimer. This hypothesis predicted that deleting the N-terminus should destabilize the dimer and thus increase LRRK1 kinase activity. We tested this by measuring phosphorylation of Rab7a in cells. We expressed one of three constructs in 293T cells: full-length LRRK1, a 25-residue N-terminal deletion (LRRK1(Δ1–25)), and a 48-residue deletion (LRRK1(Δ1–48)). In agreement with our hypothesis, both deletions resulted in a ~50% increase in phosphorylation of Rab7a compared to full-length LRRK1 (Fig. 5d and Extended Data Fig. 3). This is comparable to what we observed with the hyperactive kinase mutant LRRK1[K746G] (equivalent to the PD-linked R1441G mutation in LRRK2) (Fig. 5d and Extended Data Fig. 3). The difference in activity between LRRK1(Δ1–25) and LRRK1(Δ1–48) was not statistically significant, suggesting that the first 25 residues are involved in stabilizing the LRRK1 dimer. To obtain a measure of the $K_d$ for dimerization, we analyzed wild-type (WT), full-length LRRK1 using mass photometry. Based on our analysis, we estimate the $K_d$ for dimerization to be 0.6 μM (Extended Data Fig. 6). This value is almost nine times lower than the concentration we used to prepare cryo-EM grids with full-length LRRK1, in agreement with our observation that most of the protein was found in dimers. Although LRRK1 expression levels may be low, depending on cell type[23], LRRK1 has been shown to accumulate near the plasma membrane[24], where the locally higher LRRK1 concentrations may drive dimerization.

Given these results, we used AlphaFold multimer to model a dimer of the first 25 residues in LRRK1 (Extended Data Fig. 7a). Although the per-residue confidence values are low for the entire 25 residues (as would be expected from AlphaFold's prediction that the first 48 residues of LRRK1 are disordered), the model shows an interface formed by a short antiparallel β-sheet involving residues 11–15. Interestingly, these residues (and the next one) are the most highly conserved motif at the N-terminus of vertebrate LRRK1s (Extended Data Fig. 7b).

## LRRK1's dimer is stabilized by a loop in the WD40 domain

We noticed a large density in the LRRK1 dimer adjacent to the C-terminal helices and connecting the two WD40 domains (Fig. 5e and Supplementary Video 1); this density was weak and seen only when the map was displayed at lower threshold. We wondered if this was an artifact due to dynamic masking during processing in cryoSPARC[25] (https://cryosparc.com/), or to the two-fold symmetry applied to the map of the dimer. We reprocessed the data either using a mask that excluded the region where the density had appeared, or without applying symmetry. In both cases, the unaccounted-for density persisted. We thus wondered if the long LRRK1-specific loop in the WD40 domain (residues 1,791–1,907; Fig. 5f), which we had not been able to model, could be involved in forming the density, and in stabilizing the autoinhibited LRRK1 dimer.

We engineered a deletion of most of this loop, LRRK1(Δ1,798–1,885), predicted (by AlphaFold modeling) to maintain proper folding of the WD40 domain, and measured phosphorylation of Rab7a in 293T cells expressing either full-length LRRK1, or the LRRK1(Δ1,798–1,885) construct. In agreement with our hypothesis, deletion of the WD40 loop resulted in a significant increase in Rab7a phosphorylation in cells (Fig. 5g and Extended Data Fig. 3).

## A loop from COR-B inhibits LRRK1's kinase

Our initial model for the kinase domain of LRRK1 showed density in the back pocket of the kinase that was not accounted for by the model. The density was located where Y1410 from the DYG motif would dock in the DYG 'in', or active, conformation. Symmetry expansion and focused refinement of the dimer dataset showed that a loop from the COR-B domain (residues 1,048–1,082), which is predicted by AlphaFold to be entirely disordered (Extended Data Fig. 8a,b), threads into the kinase domain active site (Fig. 6a and Supplementary Video 1). The equivalent loop in LRRK2 is half as long and does not extend towards the active site (Extended Data Fig. 8c). The side chain of F1065, at the tip of the loop, sits inside the back pocket of the kinase (Fig. 6b), occupying the position of Y1410 in the DYG-in conformation. A similar 'plugging' of the kinase back pocket was observed in the DDR1 kinase (Extended Data Fig. 8d–f)[26]. This suggests that the COR-B loop is an autoinhibitory element in LRRK1. We tested this by measuring Rab7a phosphorylation in 293T cells expressing either WT LRRK1 or LRRK1(F1065A), which we expected would at least partially relieve the autoinhibition. In agreement with this, the F1065A mutation led to a two-fold increase in the level of Rab7a phosphorylation in cells, comparable to that observed with the hyperactive K746G mutant (Fig. 6c and Extended Data Fig. 3).

LRRK1 contains several consensus sites for phosphorylation by Protein Kinase C (PKC)[24]. Three of these—S1064, T1074 and S1075—are found in the autoinhibitory COR-B loop (Fig. 6d), and their phosphorylation significantly increases LRRK1's kinase activity[24]. In addition, preventing phosphorylation (by mutating the residues to alanine) reduces Rab7a phosphorylation in cells, while phosphomimetic mutations (to glutamate) increase it, although to a lesser extent than phosphorylation[24]. Our structure provides a mechanistic explanation for this activation: phosphorylation of these residues disrupts the loop, which is nestled against the kinase domain, releasing F1065 from the kinase's back pocket and allowing the DYG motif to adopt the active 'in' conformation. Based on this model, one might expect that combining the F1065A mutation and phosphomimetic mutations in the COR-B loop (S1064E/T1074E/ S1075) would not result in further activation, as the F1065A mutation already removes the autoinhibitory interaction. In agreement with this, LRRK1 carrying all four mutations resulted in the same increase in Rab7a phosphorylation in cells as seen with either F1065A or the phosphomimetic mutations (Fig. 6c,e,f and Extended Data Fig. 3).

Our work revealed two separate autoinhibitory mechanisms in LRRK1: (1) autoinhibition by the COR-B loop, present in both the monomer and dimer structures, and (2) steric autoinhibition of the kinase by the ANK domain, which occurs in *trans* and is dependent

on dimerization. Given the seemingly independent nature of these mechanisms, we wondered if their effects would be additive. To test this, we introduced the phosphomimetic mutations in the context of the N-terminal deletions: LRRK1(Δ1–25)(S1064E/S1074E/T1075E) and LRRK1(Δ1–48)(S1064E/S1074E/T1075E). As shown previously[24], Rab7a phosphorylation in 293T cells expressing full-length LRRK1 carrying the triple phosphomimetic mutations increased by a factor of 2 relative to WT LRRK1 (Fig. 6f and Extended Data Fig. 3). Combining these mutations with either the 1–25 or 1–48 N-terminal truncation of LRRK1 did not result in a statistically significant increase in Rab7a phosphorylation in cells (Fig. 6f and Extended Data Fig. 3). It remains to be seen whether the phosphomimetic mutants in the COR-B loop disrupt dimerization on their own, thus negating the effect of the N-terminal deletions.

## An evolutionary analysis of LRRK1 and LRRK2

Structural information on LRRK2 has built up over the last few years[14–17]. The data we presented here on LRRK1 allow us to establish the structural signatures that define these two proteins. We set out to analyze the conservation of these features throughout evolution to understand which ones are most likely to be tied to LRRK1- or LRRK2-specific biological functions. We expect that this information will shed light on the etiology of PD and bone diseases.

We began by evaluating the evolutionary origin and phylogenetic distribution of LRRK proteins, defined as those with 40% or higher sequence coverage relative to human LRRK1 or LRRK2, to ensure complete coverage of the ROC, COR-A, COR-B and kinase domains (Methods). We found that LRRK proteins are present in a wide range of metazoan species, and that related proteins are present in amoeba. Phylogenetic analyses of these proteins revealed five distinct and well-supported LRRK clades, with amoeba proteins forming a single clade and the remaining four clades containing only metazoan proteins (Fig. 7a and Extended Data Fig. 9), as has been observed previously[27]. Notably, arthropod and nematode LRRK proteins, which are annotated as either LRRK1 or LRRK2, are in fact found in a clade (labeled LRRK3 in Fig. 7a) that is distinct from vertebrate LRRK1 and LRRK2.

We next asked when characteristic features of LRRK1 and LRRK2 arose during metazoan LRRK protein family evolution, focusing our evolutionary analysis on four structural features that distinguish LRRK1 from LRRK2: (1) the LRRK1-specific WD40 loop involved in autoinhibition, (2) the LRRK1-specific COR-B loop involved in autoinhibition, (3) the three LRRK2-specific basic patches found in the ROC domain that mediate microtubule binding[17], and (4) the length of the αC helix in the kinase's N-lobe, which is much longer in LRRK1 than in LRRK2 (Fig. 7c,d). We were not able to analyze two other features—the presence of the COR-B:COR-B and WD40:WD40 dimerization interfaces in LRRK2 that are required for the formation of the microtubule-associated filaments, and the differences in length in the C-terminal helix that emerges from the WD40 domain—due to the fact that the protein alignment in these regions was not of sufficient quality to confidently infer relatedness. Consistent with the role that the WD40 loop plays in LRRK1 regulation,

**Fig. 4 | Comparison of dimeric LRRK1 and LRRK2. a,b**, Models of LRRK1 (**a**) and LRRK2 (**b**) shown in surface representation. The models are shown with the bottom-right monomer in the same orientation to highlight the differences in the architecture of the dimer. The different interfaces involved in forming the LRRK1 (**a**) and LRRK2 (**b**) dimers are indicated. In the case of LRRK1, panel letters next to the interface labels refer to the detailed views shown in this figure. Cartoons of the dimers are shown below the models. **c**, Close-up of symmetric interface formed by the ANK domains. **d**, The interface formed by the ANK domain of one monomer and the LRR domain of the other monomer brings together surfaces of complementary charge. **e**, The kinase C-lobe of one monomer interacts both with the LRR domain of the same monomer and the ANK domain of the other monomer. Along with the interaction in **g**, this anchors the ANK domain on top of the kinase, where it blocks access to the active site. **f**, Symmetric interaction between the N-lobes of the kinases. This is the only dimeric interface involving only a domain in the C-terminal half of LRRK1 (RCKW). **g**, The N-lobe of the kinase of one monomer interacts with the ANK domain of the other monomer. Along with the interaction shown in **e**, this anchors the ANK domain on top of the kinase, where it blocks access to the active site. In **c**–**g**, the insets highlight the area of the structure shown in the main panel. Except for **d**, all other panels show the LRRK1 model inside the cryo-EM map. **h**, The cryo-EM map of the LRRK1 dimer, colored by domains, is shown in the same orientation as the model in **a**. **i**, Rotated view of the dimer map showing how the kinase active site is buried. **j,k**, An additional rotation of the dimer (**j**) and clipping of the density in front (**k**) highlights how the kinase from one monomer is buried under the ANK domain from the other monomer.

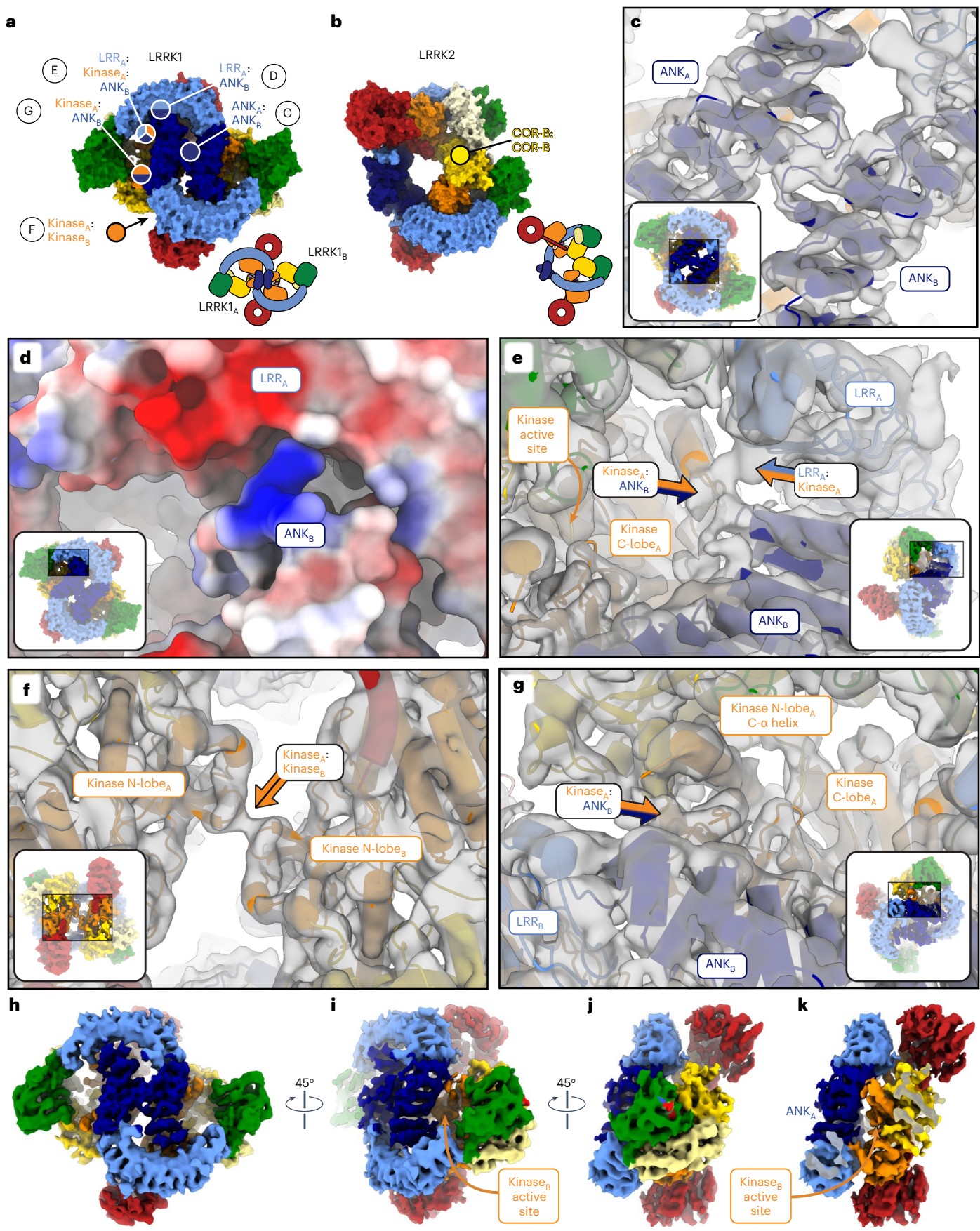

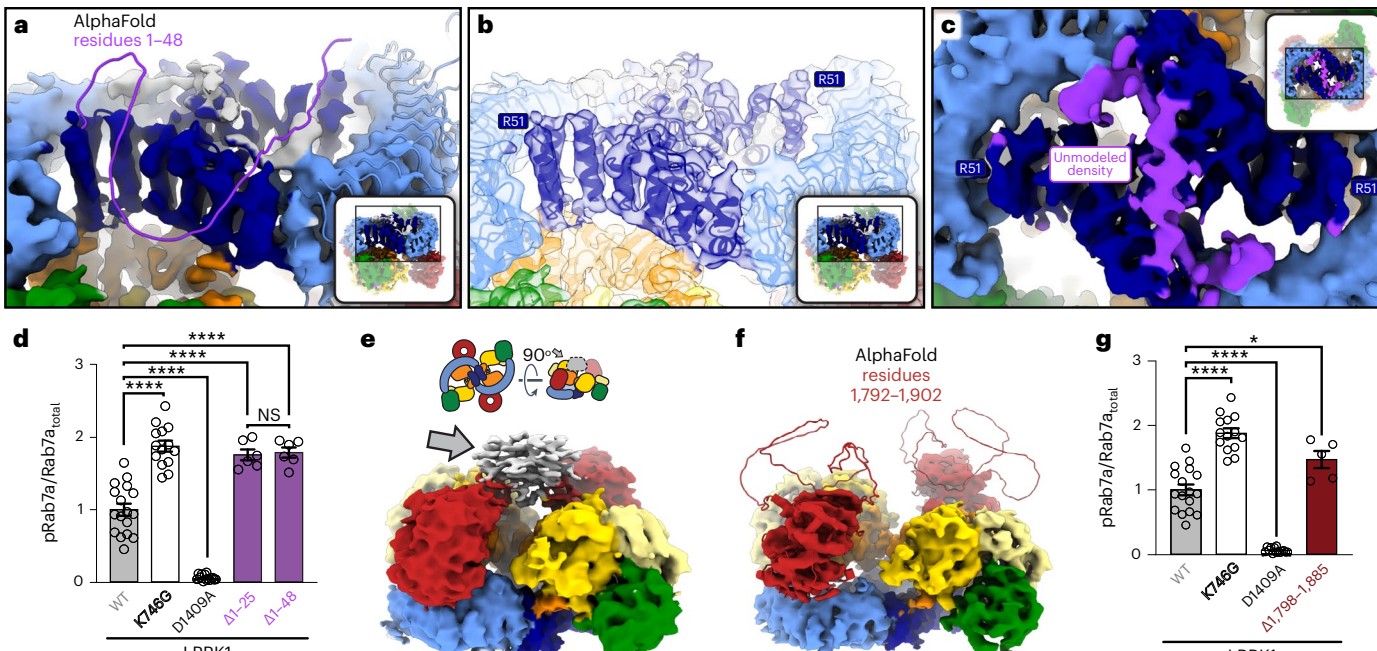

**Fig. 5 | Disordered loops in LRRK1's ANK and WD40 domains help stabilize the autoinhibited dimer. a**, The ANK and LRR domains of the AlphaFold model of human LRRK1 (Q38SD2) shown docked into our cryo-EM map of LRRK1's dimer. The N-terminal residues 1–48, which are unstructured in the AlphaFold model and not included in our model, are shown in purple. The insets in **a**–**c** highlight the area of the structure shown in the main panel. **b**, Same view as in **a** with our model of the LRRK1 dimer shown inside the cryo-EM map to highlight that R51 is the first residue modeled in our structure. **c**, The ANK–ANK interface in the LRRK1 dimer. The purple density corresponds to the region of the cryo-EM map unaccounted for by our current model. R51, where our model begins, is indicated. **d**, Rab7a phosphorylation in 293T cells expressing full-length LRRK1 or N-terminally truncated constructs missing the first 25 or 48 residues. LRRK1(K746G), which increases Rab7 phosphorylation in cells, and LRRK1(D1409A), which inactivates the kinase, were also tested. 293T cells were transiently transfected with the indicated plasmids encoding FLAG–LRRK1 (WT or mutant) and GFP–Rab7. Thirty-six hours post-transfection the cells were lysed, immunoblotted for phospho-Rab7 (pS72), total GFP–Rab7 and total LRRK1. The mean ± s.e.m. is

shown, ****$P < 0.0001$; NS, not significant; one-way ANOVA with Tukey's multiple comparisons test. Individual data points represent separate populations of cells obtained across at least three independent experiments (*n* values shown in EDF 3). **e**, Weak density (gray arrow) connecting the WD40 domains in our dimer maps. Cartoons above the map indicate the orientation of the maps in **e** and **f** and the location of the weak density. **f**, Cryo-EM map of the LRRK1 dimer with the WD40 domain from the AlphaFold model of LRRK1 docked in; residues 1,792–1,902 were predicted to form a disordered loop. **g**, Rab7a phosphorylation in 293T cells expressing GFP–Rab7a and full-length WT LRRK1 or a LRRK1 variant missing residues 1,798–1,885 from its WD40 domain. LRRK1(K746G) and LRRK1(D1409A) were used as controls. 293T cells were transiently transfected with the indicated plasmids encoding FLAG–LRRK1 (WT or mutant) and GFP–Rab7. Thirty-six hours post-transfection the cells were lysed, immunoblotted for phospho-Rab7 (pS72), total GFP–Rab7 and total LRRK1. The mean ± s.e.m. is shown. *$P = 0.0317$, one-way ANOVA with Tukey's multiple comparisons test. Individual data points represent separate populations of cells obtained across at least three independent experiments (*n* values shown in EDF 3).

we found that this is a conserved feature of metazoan LRRK1s not found in any other metazoan LRRKs (Fig. 7d,e and Extended Data Fig. 10). We also found that an extended (>27 residues) COR-B loop is conserved in LRRK1 proteins (Fig. 7f and Extended Data Fig. 10). Our structure of LRRK1 suggests that the COR-B loop, as defined in Fig. 7d, must be at least 26 residues long for the autoinhibitory mechanism we identified to be possible. Our analysis showed that all three basic patches that mediate microtubule binding are found only in LRRK2s from jawed vertebrates (Fig. 7g and Supplementary Fig. 1). All other LRRK2s we examined had at least the first two basic patches. Finally, the length of the αC helix in the kinase's N-lobe is a feature found in both LRRK1s and LRRK3s (Fig. 7h).

## Discussion

Our structural and functional analysis of LRRK1 has revealed that although LRRK1 and LRRK2 have a similar domain architecture and related kinase substrates, the two proteins differ in the structures of their monomers and dimers and, most dramatically, in their mechanisms of autoinhibition and activation.

### LRRK1 regulation

One of the most striking aspects of our findings is the extent to which LRRK1 appears to be more stringently autoinhibited than LRRK2.

Our current understanding of LRRK2 suggests that its activation should only require movement of the N-terminal repeats to relieve the physical blockage of the kinase's active site by the LRR. Structures and structure predictions of LRRK2 indicate that this activation could take place in the context of either a monomer or a dimer as dimerization does not bring about new properties, at least in the context of the inactive dimer form[16]. In contrast, the equivalent steric inhibition in LRRK1, mediated by its ANK repeats, is dependent on dimerization. Given that the ANK repeats are also involved in mediating dimerization, along with several other homotypic and heterotypic interactions in LRRK1, it is not easy to see how autoinhibition in LRRK1 could be relieved without disrupting the dimer. Conversely, because the kinase's active site is also inhibited by the COR-B loop, an inhibition we observed in both the monomer and dimer structures, disruption of the dimer is expected to be necessary but not sufficient for full activation. Our structure of the dimer revealed seven different interfaces that appear to stabilize it (Figs. 4 and 5), of which two were tested here: the extreme N-terminus of LRRK1, and the LRRK1-specific WD40 loop near the C-terminus. Given that in both cases we disrupted only one of the seven interfaces present in the dimer, it was to be expected that activation (measured as an increase in Rab7a phosphorylation in cells) would be relatively modest. A more systematic dissection

will be required to understand how each of the individual interfaces contributes to the stability of the dimer.

How could LRRK1 dimerization be regulated in cells? An intriguing observation is that there are multiple predicted phosphorylation sites (phosphosite.org) in most of the interfaces we identified, including the two regions we tested here. Future studies will determine whether phosphorylation of residues in these different interfaces disrupts, or stabilizes, the autoinhibited LRRK1 dimer.

The autoinhibitory COR-B loop is unique to LRRK1 and provides an additional level of regulation, present in both our monomer and dimer structures. Our structure suggests that disruption of the dimer should precede phosphorylation of the COR-B residues that lead to activation; the COR-B loop is relatively buried in the structure, surrounded by the ROC and LRR domains from the same monomer, and the ANK repeats that come from the other monomer to block the kinase (Fig. 4h–k). Future studies will also determine whether, in addition to relieving autoinhibition, phosphorylation of the COR-B loop enables new interactions, with LRRK1 and/or other partners.

Although much remains to be done to understand how LRRK proteins are regulated, it is tempting to speculate that the apparently more stringent regulation of LRRK1 may be related to the fact that there are no validated gain-of-function disease-linked mutations in LRRK1, in contrast to those in PD-linked LRRK2. LRRK1 is also expressed more broadly across tissues than LRRK2, which shows highest expression in lung and whole blood (Genotype-Tissue Expression (GTEx) portal). It may be that a hyperactive LRRK1 is more acutely deleterious than a hyperactive LRRK2 in cells. More stringent regulation would then be expected to prevent spurious activation.

## Structural signatures of LRRK1 and LRRK2

One of our goals here was to determine what structural features are unique to LRRK1 or shared between LRRK1 and LRRK2. LRRK1-specific features would shed light on LRRK1's involvement in bone diseases, while those specific to LRRK2 would shed light on its involvement in PD.

Our evolutionary analysis showed that two of the structural elements involved in LRRK1's autoinhibition—a long loop in the WD40 domain and a COR-B loop that directly inhibits the kinase—are conserved in LRRK1s but absent in LRRK2s, suggesting that autoinhibition is central to LRRK1's function. Interestingly, LRRK3s, including those found in arthropods (for example, *Drosophila*) and nematodes (for example, *Caenorhabditis*) lack the long WD40 loop but share an extended COR-B loop with LRRK1. While it is unclear what role these might play in LRRK3 function in these species, it does suggest that studies of LRRK protein function from model invertebrates should be treated with caution when extrapolating to human LRRKs. Conversely, our analysis showed that the LRRK2-specific basic patches found in its ROC domain, which mediate LRRK2's interaction with microtubules, are absent in LRRK1s but conserved in LRRK2s, although the full complement of three basic patches is found only in jawed vertebrates. We note that mutating a single basic patch (either 1 or 2) was sufficient to abolish microtubule binding in human LRRK2 (ref. 17), and that these are the basic patches conserved in all LRRK2s. Although the physiological relevance of microtubule binding by LRRK2 remains to be established, our data suggest that the functional role of the basic patches in LRRK2 is unique to LRRK2.

The last structural feature we analyzed that distinguishes LRRK1 and LRRK2 is the length of their αC helices, found in the N-lobe of the kinase. Unlike the features discussed before, for which we have shown

### Fig. 6 | A loop from the COR-B domain directly inhibits LRRK1's kinase.

**a**, Close-up of the kinase domain in the cryo-EM map of monomeric LRRK1; the yellow density corresponds to a loop from the COR-B domain that reaches the kinase active site. The inset highlights the area of the structure shown in the main panel. The dashed outline indicates the region shown in **b**. **b**, Our model of LRRK1 shown inside the cryo-EM map around the kinase's active site. The DYG motif, in its 'out' conformation, is shown. F1065, a residue in the COR-B inhibitory loop, occupies the kinase's 'back pocket', where Y1410 must dock to bring the DYG motif into its 'in', or active, conformation. **c**, Rab7a phosphorylation in cells expressing full-length LRRK1 WT or carrying a F1065A mutation. LRRK1(K746G) and LRRK1(D1409A) are the same controls used in Fig. 5. 293T cells were transiently transfected with the indicated plasmids encoding for FLAG–LRRK1 (WT or mutant) and GFP–Rab7. Thirty-six hours post-transfection the cells were lysed, immunoblotted for phospho-Rab7 (pS72), total GFP–Rab7 and total LRRK1. The mean ± s.e.m. is shown. ****P < 0.0001, one-way ANOVA. Individual data points represent separate populations of cells obtained across at least three independent experiments (n = 8, except for F1065A, where n = 3). **d**, Expanded view of the area shown in **b**, without the cryo-EM map. The three sites of PKC phosphorylation in the COR-B inhibitory loop—S1064, S1074 and T1075—are shown in addition to F1065. **e,f**, Rab7a phosphorylation in 293T cells expressing GFP–Rab7a and full-length WT LRRK1 or LRRK1 carrying a combination of the F1065A mutation with three phosphomimetic mutations (S1064E/S1074E/ T1075E) in the COR-B inhibitory loop (**e**) or truncated (Δ1–48 and Δ1–25) versions of LRRK1 with or without the phosphomimetic mutations (S1064E/S1074E/ T1075E) in the COR-B inhibitory loop (**f**). The triple phosphomimetic mutant is abbreviated as 'S/T → E' in the graphs. LRRK1(K746G) and LRRK1(D1409A) were used as controls. 293T cells were transiently transfected with the indicated plasmids encoding for FLAG–LRRK1 (WT or mutant) and GFP–Rab7. Thirty-six hours post-transfection the cells were lysed, immunoblotted for phospho-Rab7 (pS72), total GFP–Rab7 and total LRRK1. The mean ± s.e.m. is shown. ****P < 0.0001 (P = 0.6730 for Δ1–25 versus Δ1–48). One-way ANOVA. Individual data points represent separate populations of cells obtained across at least three independent experiments (n = 3 (**e**) and 6 (**f**)).

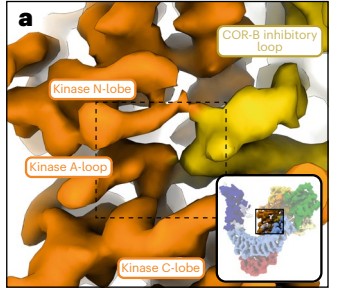
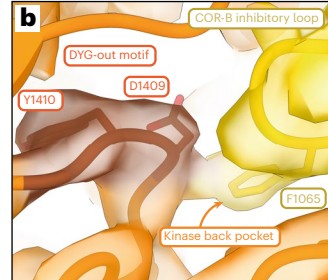
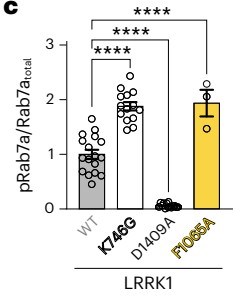
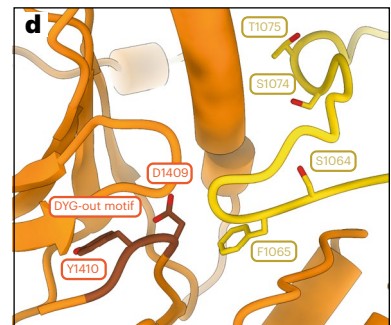
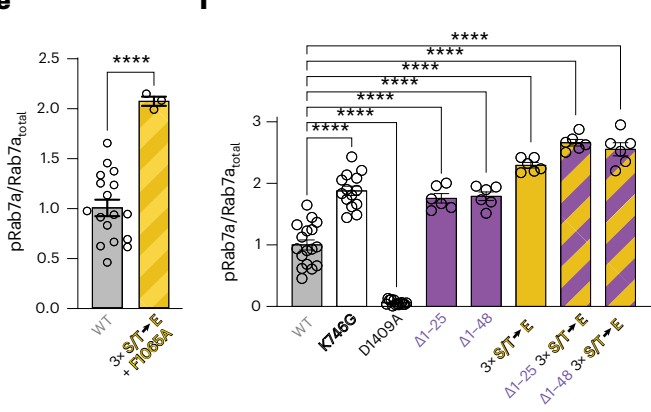

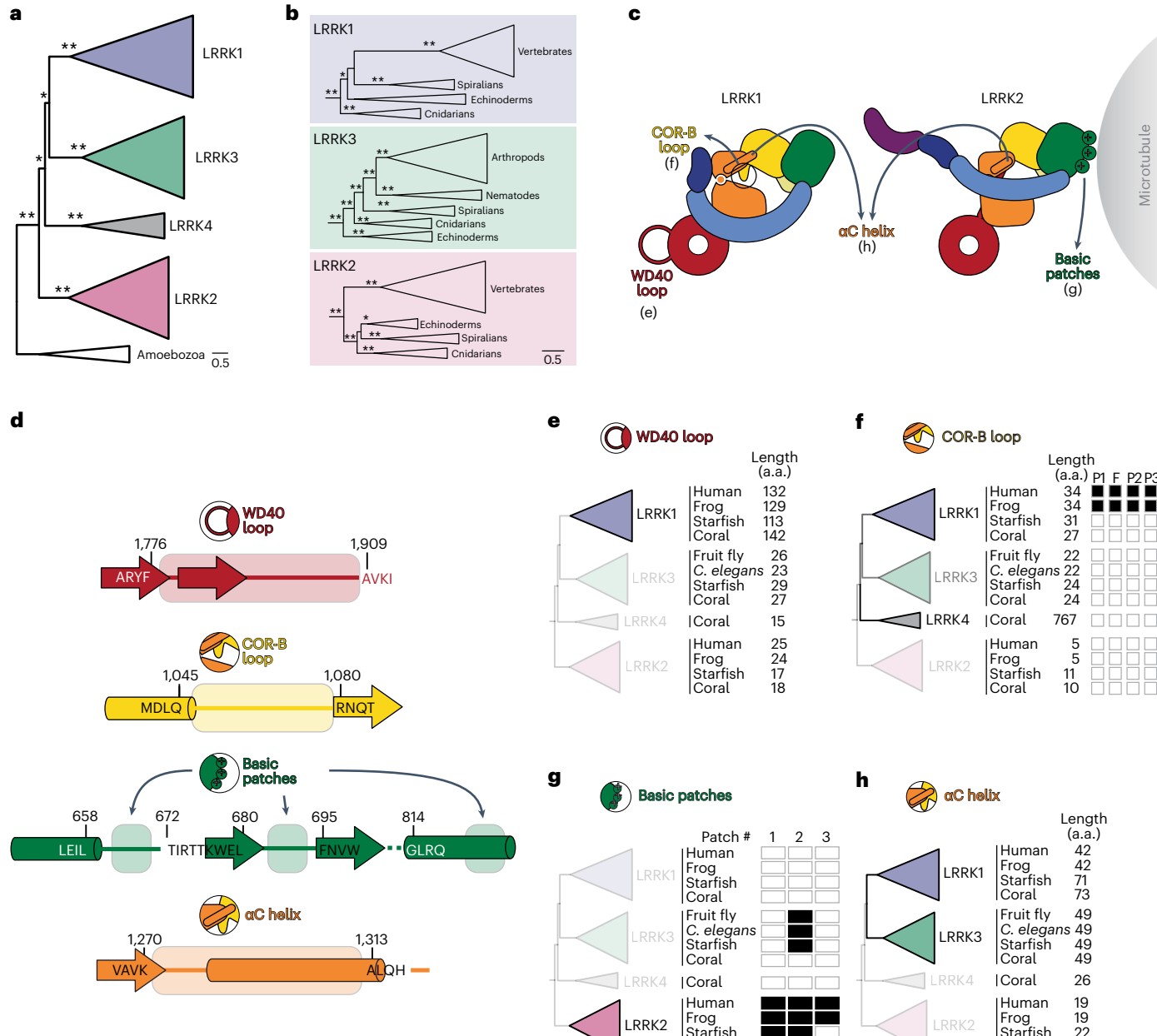

**Fig. 7 | Evolution of structural motifs in LRRK proteins. a**, Maximum likelihood phylogenetic tree of LRRK protein homologs (for complete listing of proteins, protein alignment and complete phylogenetic tree, respectively, see Supplementary Datasets 1–3). Shaded clades are LRRK proteins found in metazoans, using nomenclature proposed in ref. 27. The tree is rooted on the only non-metazoan LRRK homologs, found in Amoebozoa, but their validity as the ancestor to metazoan LRRK proteins is not well established[28]. Asterisks indicate bootstrap branch support (*>75% support, **100% support). **b**, Expanded views of the phylogenetic tree shown in **a**, highlighting major metazoan clades that contain members of LRRK1, LRRK2 and LRRK3. Asterisks indicate bootstrap branch support as in **a**. **c**, Cartoons of LRRK1 and LRRK2 showing structural features whose conservation was analyzed here. The panels where results are presented for each feature are indicated. **d**, Schematic of the boundaries used for determining whether a feature shown in **c** is present in a given LRRK. Amino acid (a.a.) sequences and numbers are shown for regions flanking the sequences whose lengths (**e,f,h**) or where the presence of key residues in the

COR-B autoinhibitory loop (**f**), or basic patches (**g**) were measured in our analysis. **e**, Representative proteins from each metazoan LRRK protein family were sampled from vertebrates (human (*Homo sapiens*) and frog (*Xenopus laevis*)), echinoderms (starfish: *Asterias rubens*), cnidarians (coral: *Dendronephthya gigantea*), arthropods (fruit fly: *Drosophila sechellia*) and nematodes (*Caenorhabditis elegans*). The length of the WD40 loop, as measured between the well-aligning motifs shown in **d**, is shown next to each homolog. **f**, As in **e**, except measuring the length of the COR-B loop between the well-aligning motifs shown in **d**. Filled boxes indicate the presence of key residues involved in autoinhibition and activation—the Phe that docks into the kinase back pocket ('F'), and the three phosphorylation sites ('P1–P3'). **g**, As in **e**, except querying for the presence of basic patches. Filled boxes indicate the presence of a basic patch as defined by three basic amino acids in a stretch of four residues between the regions defined in **d**. **h**, As in **e**, except measuring the length of the region containing the αC helix between the well-aligning motifs shown in **d**.

specific functions, we do not yet understand the mechanistic implications of the length of the αC helix, though our evolutionary analysis indicated that a longer helix is a signature of LRRK1s and LRRK3s.

Our structures of LRRK1 suggested that the more extensive interaction between a longer αC helix and a hydrophobic pocket in the COR-B domain may rigidify LRRK1 relative to LRRK2 (Extended Data Fig. 2).

Future studies will determine whether LRRK1 and LRRK2 have different dynamics despite their similar domain architecture, and whether those are related to their function and regulation. One possibility is that increased rigidity in LRRK1 facilitates the formation of its auto-inhibited dimer.

## Online content

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

## Methods

### Cloning and mutagenesis

For baculovirus expression, the DNA coding for LRRK1 was codon optimized for *Spodoptera frugipera* (Sf9) cells and synthesized by Epoch Life Science. The DNA was cloned via Gibson assembly into the pKL baculoviral expression vector, with an N-terminal His$_6$-Z-tag and a TEV protease cleavage site. LRRK1 variants were generated using Q5 site-directed mutagenesis (New England Biolabs, NEB). The pKL plasmid was used for the generation of recombinant Baculoviruses using the Bac-to-Bac expression system (Invitrogen).

For mammalian expression, pcDNA5–FRT–TO–LRRK1 from Medical Research Council–Protein Phosphorylation and Ubiquitylation Unit (https://www.ppu.mrc.ac.uk/) was used. The various LRRK1 mutants were generated using QuikChange site-directed mutagenesis (Agilent), or Q5 site-directed mutagenesis (NEB) following the manufacturer's instructions. Plasmid design was performed using SnapGene software (Insightful Science; snapgene.com), and all plasmids were sequence verified. Enhanced green fluorescent protein (EGFP)–Rab7 was obtained from Addgene (#12605).

Oligonucleotide sequences are listed in a separate spreadsheet ('Oligonucleotides' in Supplementary Table 1).

### LRRK1 expression and purification

N-terminally tagged His$_6$–Z–TEV–LRRK1(FL) was expressed in Sf9 insect cells. Insect cells were infected with baculovirus and grown at 27 °C for 3 days. Cells were collected and cell pellets were resuspended in lysis buffer (50 mM HEPES pH 7.4, 500 mM NaCl, 20 mM imidazole, 0.5 mM tris(2-carboxyethyl)phosphine (TCEP), 5% glycerol, 5 mM MgCl$_2$, 20 μM GDP, 0.5 mM Pefabloc and cOmplete EDTA-free protease inhibitor cocktail (Roche). Cells were lysed using a Dounce homogenizer and clarified by centrifugation. The supernatant was incubated for 1 h with Ni-NTA agarose beads (Qiagen) equilibrated in lysis buffer. Beads were applied to a gravity column where they were extensively washed with lysis buffer, followed by elution in lysis buffer containing 300 mM imidazole. The eluted protein was diluted to 250 mM NaCl and loaded onto a SP Sepharose column (Cytiva) equilibrated in buffer (50 mM HEPES pH 7.4, 250 mM NaCl, 0.5 mM TCEP, 5% glycerol, 5 mM MgCl$_2$, 20 μM GDP and 0.5 mM Pefabloc). The protein was eluted using a 250 mM to 2.5 M NaCl gradient. Fractions containing LRRK1(FL) were pooled, diluted to ~500 mM NaCl, and incubated with TEV protease overnight at 4 °C. The protein was concentrated and put directly over a S200 size exclusion column (Cytiva) equilibrated in storage buffer (50 mM HEPES pH 7.4, 150 mM NaCl, 0.5 mM TCEP, 5% glycerol, 5 mM MgCl$_2$ and 20 μM GDP). The protein was concentrated to ~5–6 μM and flash frozen in liquid nitrogen for storage. N-terminally tagged His$_6$–Z–TEV–LRRK1$^{20-2015}$ was purified using the same protocol.

### Mass photometry

Full-length LRRK1 was diluted to 100 nM in assay buffer (20 mM HEPES pH 7.4, 150 mM NaCl, 2.5 mM MgCl$_2$, 1 mM GDP, 0.5 mM TCEP and 0.5% glycerol) and incubated for 30 min at room temperature. The samples were then analyzed in a Refeyn TwoMP mass photometer. Three independent measurements were performed with data collected for 60 s each. Gaussian curves were fitted in the histograms for determination of the dimerization grade. The law of mass action was applied to estimate the $K_d$ for dimerization.

### Cryo-electron microscopy

**Sample preparation.** For LRRK1(Δ1–19), grid samples contained 2 μM LRRK1(Δ1–19), 3 μM Rab7a, purified as previously described[17], 1 mM GTP and 1 mM AMPcPP. For LRRK1(FL), 5.2 μM LRRK1(FL) was spiked with a final concentration of 0.06% brij-35 directly before vitrification. For both samples, 4 μl of sample was applied to a freshly plasma-cleaned Ultrafoil grid (Electron Microscopy Sciences). Samples were blotted

for 4 s using a blot force of 20 followed by vitrification using a Vitrobot Mark IV (FEI) set to 4 °C and 100% humidity.

**Data collection.** Cryo-EM data were collected on a Talos Arctica (FEI) operated at 200 kV and equipped with a K2 Summit direct electron detector (Gatan). Leginon[29] (http://emg.nysbc.org/redmine/projects/leginon/wiki/Leginon_Homepage) was used for automated data collection. For the LRRK1(Δ1–19) dataset, we collected 1,310 movies at a nominal magnification of 36,000× and object pixel size of 1.16 Å. Movies were dose-fractionated into 200 ms frames for a total exposure of 12 s with a dose rate of ~5.1 electrons Å$^{-2}$ s$^{-1}$. For the LRRK1(FL) sample, multiple datasets were collected and combined as the addition of detergent resulted in very few particles per image. Movies were collected with the same parameters as the LRRK1(Δ1–19) dataset.

**LRRK1(Δ1–19) monomer reconstruction.** cryoSPARC Live[25] was used to align movie frames (patch motion correction) and estimate the CTF (patch CTF estimation). Micrographs with a CTF worse than 6 Å were removed. Particles were picked using crYOLO[30] (http://sphire.mpg.de/wiki/doku.php?id=pipeline:window:cryolo) with a previously trained model on the dose-weighted images. Particles were extracted with a downsampled pixel size of 4.64 Å per pixel and used in multiple rounds of 2D classification in Relion 3.0 (ref. 31) (http://www2.mrc-lmb.cam.ac.uk/relion). Particles were extracted to 1.16 Å per pixel and all subsequent processing was done in cryoSPARC[25]. Particles were subjected to ab initio reconstruction, and the best two classes were combined and used in non-uniform refinement. To further separate out heterogeneity, another round of ab initio reconstruction was performed. Heterogeneous refinement was then done to separate particles with and without the LRR domain. 2D classification was carried out on each resulting subset to remove lingering dimer particles or recover good particles. Particles were then combined and put through a last round of non-uniform refinement for a final resolution of 3.7 Å.

**LRRK1(FL) dimer reconstruction.** cryoSPARC Live[25] was used to align movie frames (patch motion correction) and estimate the CTF (patch CTF estimation). Particles were picked using crYOLO[30] trained with manual particle picks. Particles were extracted with a downsampled pixel size of 4.11 Å per pixel and used in multiple rounds of 2D classification implemented in cryoSPARC[25]. Good particles were subjected to ab initio reconstruction. The best two classes were combined, and the particles were used in heterogeneous refinement. Final particles were taken and used in non-uniform refinement with C2 symmetry and optimized per-group CTF params enabled for a final resolution of 4.6 Å. Symmetry expansion followed by local refinement resulted in a monomer structure of 3.6 Å.

**Model building for monomeric LRRK1(Δ1–19).** The AlphaFold[21] model of human LRRK1, Q38SD2, was docked into the LRRK1(Δ1–19) map. This model has the kinase modeled in the closed conformation, and so we used Phenix[32] (https://www.phenix-online.org/) real space refine to rigid body fit each domain into the map. Discrepancies in the AlphaFold model were manually corrected in COOT[33]. To aid in model building, local refinements with masks around the KW domains (EMDB 27816), RCKW domains (EMDB 27814) and RCK domains (EMDB 27815) were performed. These maps only minimally improved the local resolution (~0.1–0.3 Å), and therefore specific models for these maps were not included. Model refinement was done using a combination of Phenix real space refine[32] and Rosetta Relax (version 3.13)[34] (https://www.rosettacommons.org/home).

**Model building for dimeric LRRK1(FL).** The monomer model for LRRK1$^{RCKW}$ and the ANK–LRR domains from AlphaFold[21] model Q38SD2 were docked into the dimer symmetry expansion map and rigid body fit using Phenix[32] real space refine. The model was iteratively built in

COOT[33] (http://www2.mrc-lmb.cam.ac.uk/personal/pemsley/coot/) and refined using Phenix[32] real space refine and Rosetta Relax (version 3.13)[34].

**Visualization.** ChimeraX[35] was used for visualizing structures and to prepare figures.

**Cell culture and transfection.** 293T cells were obtained through ATCC (CRL-3216). Cells were cultured in DMEM growth medium (Gibco) supplemented with 10% fetal bovine serum (Gibco) and 1× penicillin/streptomycin solution (Gibco). Cells were plated on six-well dishes (200,000 cells per well) 24 h before transfection. Each culture well was transfected with 1 μg of FLAG–LRRK1 construct and 500 ng of EGFP–Rab7 using polyethylenimine (Polysciences) and cultured at 37 °C with 5% CO$_2$ for 36 h before collection.

**Western blot analysis and antibodies.** For western blot quantification of LRRK1 protein expression and Rab7 phosphorylation, cells were collected by scraping, rinsed with ice-cold 1× phosphate-buffered saline, pH 7.4 and lysed on ice in RIPA buffer (50 nM Tris pH 7.5, 150 mM NaCl, 0.2% Triton X-100 and 0.1% sodium dodecyl sulfate, with cOmplete EDTA-free protease inhibitor cocktail (Sigma-Aldrich) and PhoStop phosphatase inhibitor (Sigma-Aldrich)). Lysates were rotated for 15 min at 4 °C and clarified by centrifugation at maximum speed in a 4 °C microcentrifuge for 15 min. Supernatants were then boiled for 10 min in sodium dodecyl sulfate buffer. Replicates were performed on independently transfected cultures.

Lysates were run on 4–12% polyacrylamide gels (NuPage, Invitrogen) for 50 min at 180 V and transferred to polyvinylidene difluoride (Immobilon-FL, EMD Millipore) for 4 h at 200 mA constant current. Blots were rinsed briefly in MilliQ water and dried at room temperature for at least 30 min. Membranes were briefly reactivated with methanol and blocked for 1 h at room temperature in 5% milk (w/v) in TBS. Antibodies were diluted in 1% milk in TBS with 0.1% Tween-20 (TBST). Primary antibodies used for immunoblots were as follows: mouse anti-green fluorescent protein (GFP) (Santa Cruz, 1:2,500 dilution), rabbit anti-LRRK1 (Abcam, 1:500 dilution), rabbit anti-GAPDH (Cell Signaling Technology, 1:3,000 dilution) and rabbit anti-phospho-S72-Rab7A (MJF-38, 1:1,000 dilution). Secondary antibodies (1:10,000) used for western blots were IRDye 680RD Goat anti-Mouse and IRDye 780RD Goat anti-Rabbit (Li-COR). Primary antibodies were incubated overnight at 4 °C, and secondary antibodies were incubated at room temperature for 1 h. For quantification, blots were imaged on an Odyssey CLx Imaging System (Li-COR) controlled by Imaging Studio software (version 5.2) (Li-COR), and intensity of bands quantified using Image Studio Lite software (version 5.2; https://www.licor.com/bio/image-studio-lite/). All statistical analyses were performed in GraphPad Prism (9.3.1; GraphPad Software) (http://www.graphpad.com/). During analysis, all other bands in a lane were normalized to the GAPDH band followed by dividing the phospho-Rab7 signal by the GFP–Rab7 signal. The results of this analysis from all mutants were compiled and normalized to the mean of WT and analyzed by analysis of variance (ANOVA) followed by Tukey's multiple comparisons test.

**Phylogenetic analyses.** Human LRRK1 (accession NP_078928.3) and LRRK2 (accession NP_940980.4) were used as a BLASTp[36] (http://blast.ncbi.nlm.nih.gov/Blast.cgi?PROGRAM=blastp&PAGE_TYPE=BlastSearch&LINK_LOC=blasthome) search query against the Reference Sequence (RefSeq; http://www.ncbi.nlm.nih.gov/RefSeq/) protein database with an e-value cutoff of $1 \times 10^{-20}$ and a query coverage cutoff of 40%. Resulting sequences were aligned using Clustal Omega[37] (http://www.ebi.ac.uk/Tools/msa/clustalo/), and duplicate and poorly aligning sequences were removed. To reduce the number of nearly identical sequences, sequences with >80% sequence identity

were reduced to a single unique sequence using CD-HIT[38] (https://sites.google.com/view/cd-hit) with a 0.8 sequence identity cutoff. The resulting 273 sequences (accession numbers and species names found in Supplementary Dataset 1) were realigned with Clustal Omega using two rounds of iteration to optimize the alignment throughout the sequences. The resulting alignment is found in Supplementary Dataset 2. To generate maximum likelihood phylogenetic trees of LRRK proteins, IQ-TREE[39] (http://www.iqtree.org/) phylogenies were generated using the '-bb 1000 -alrt 1000' commands for generation of 1,000 ultrafast bootstrap[40] and SH-aLRT support values. The best substitution model (JTT + F + I + G4) was determined by ModelFinder[41] using the '-m AUTO' command. To confirm that the phylogenetic inferences were not influenced by regions of LRRK proteins that are not well conserved across all family members, the ROC–COR-A region (corresponding to human LRRK1 residues 632–995) and kinase domain region (corresponding to human LRRK1 residues 1,229–1,534) were extracted from the alignment and concatenated and used as input for IQ-TREE. The resulting phylogeny, using the best substitution model (Q.insect +I + G4), shows similarly strong support values in major branches of the phylogeny (Extended Data Fig. 9). Complete phylogenetic trees, with support values, can be found in Supplementary Dataset 3.

To determine length of WD40 and COR-B loops, and the aC helix region, well-aligning regions of the alignment, which often corresponded to ordered regions of the LRRK1 and LRRK2 structures, were used as boundaries elements to count the number of intervening residues. Boundary amino acid sequences and residue numbers are shown in Fig. 7d. For the COR-B loop in cnidarian LRRK4, the automated alignment did not identify the well-conserved WxxGφxφ C-terminal boundary element because it is 200+ residues farther from the N-terminal boundary element in cnidarian LRRK4s than in any other LRRK proteins. To measure the loop length, this well conserved WxxGφxφ motif was manually identified in cnidarian LRRK4s and used to count the intervening COR-B 'loop' region. To determine the presence of basic patches, well-aligning boundary elements flanking each LRRK2 basic patch were identified as described above. For each sequence shown in Fig. 7g, intervening sequences between the indicated boundary elements were manually searched for any occurrence of three basic residues within a four-residue window.

To identify the most conserved residues in the first 40 residues of vertebrate LRRK1s residues 1–40 of human LRRK1 were used as a search query against the RefSeq database with an e-value cutoff of 0.05. All resulting 1455 sequences were annotated as vertebrate LRRK1s. Sequences were downloaded and aligned using Clustal Omega with two iterations of refinement, and the alignment region corresponding to human residues 1–25 was extracted. Poorly aligning sequences and identical sequences were removed. The remaining sequences were realigned with Clustal Omega with two iterations of refinement. The resulting alignment is shown in Extended Data Fig. 7.

Consensus logos and alignment visualization for basic patch 3 and the COR-B loop region were generated using Geneious Prime 2022.1.1 (https://www.geneious.com).

**GTEx portal.** The GTEx Project was supported by the Common Fund of the Office of the Director of the National Institutes of Health, and by National Cancer Institute, National Human Genome Research Institute, National Heart, Lung, and Blood Institute, National Institute on Drug Abuse, National Institute of Mental Health and National Institute of Neurological Disorders and Stroke. The data used for the analyses described in this manuscript were obtained from the GTEx Portal on 28 January 2023.

**Reporting summary**
Further information on research design is available in the Nature Portfolio Reporting Summary linked to this article.

## Data availability

Cryo-EM maps of the LRRK1 monomer (LRRK1(Δ1–19)) (EMD-27813), the LRRK1 dimer (LRRK1(FL)) (EMD-27817), a symmetry expanded monomer from the dimer map (EMD-27818) and local refinements of the monomer structure−around KW (EMD-27816), RCKW (EMD-27814) and RCK (EMD-27815)−were all deposited in the EM Data Bank. Models of the LRRK1 monomer (PDB: 8E04), LRRK1 dimer (PDB: 8E05) and symmetry expanded monomer from the dimer structure (PDB: 8E06) were all deposited in the Protein Data Bank. Source data are provided with this paper.

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

## Acknowledgements

This research was funded in part by Aligning Science Across Parkinson's (grant number ASAP-000519) (S.K., S.L.R.-P. and A.E.L.) through the Michael J. Fox Foundation for Parkinson's Research (MJFF) (S.K., S.L.R.-P. and A.E.L.). The work was also funded by the Howard Hughes Medical Institute (where S.L.R-P. is an investigator); the Michael J. Fox Foundation (grant number 18321 to A.E.L. and S.L.R.-P.); the National Institutes of Health (R35GM133633 to M.D.D.); the Burroughs Wellcome Fund (grant number 1021386 to M.D.D.); and the National Institutes of Health (R35GM133633 to M.D.D.). We thank D. Alessi for sharing unpublished data on the phosphorylation sites in LRRK1's COR-B domain. We also thank the UC San Diego Cryo-EM Facility, the Nikon Imaging Center at UC San Diego, and the UC San Diego Physics Computing Facility for IT support. For the purpose of open access, the authors have applied a CC BY 4.0 public copyright license to all author accepted manuscripts arising from this submission.

## Author contributions

J.M.R. and Y.X.L. performed all the cryo-EM work. S.M., D.C., J.M.R. and Y.X.L. prepared samples for structural studies. A.M.D. and R.G.A. carried out all cell-based assays. E.J.F. and M.D.D. carried out the evolutionary analyses. S.K., M.D.D., S.L.R.-P. and A.E.L. supervised the work. J.M.R., S.L.R.-P., E.J.F., M.D.D. and A.E.L. wrote the paper, and J.M.R., A.M.D., R.G.A., S.M., M.D.D., S.L.R.-P. and A.E.L. edited it.

## Competing interests

S.L.R.-P. is a consultant for Schrödinger, Inc. The other authors declare no competing interests.

## Additional information

**Extended data** is available for this paper at https://doi.org/10.1038/s41594-023-01109-1.

**Correspondence and requests for materials** should be addressed to Samara L. Reck-Peterson or Andres E. Leschziner.

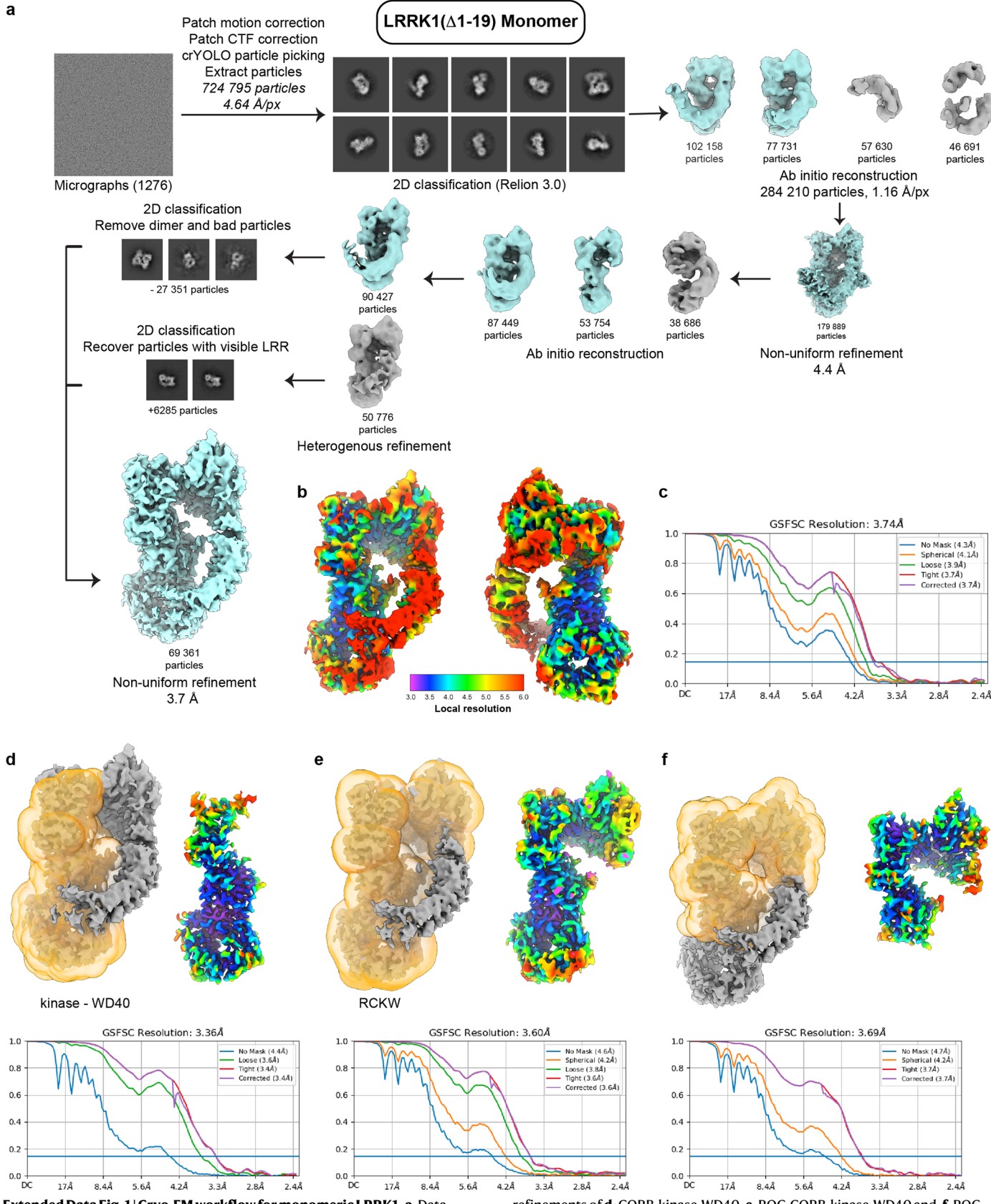

**Extended Data Fig. 1 | Cryo-EM workflow for monomeric LRRK1. a,** Data processing and reconstruction of monomeric LRRK1. **b,** Map of monomeric LRRK1 colored by local resolution. **c,** FSC plot for monomeric LRRK1. Local refinements of **d,** CORB-kinase-WD40, **e,** ROC-CORB-kinase-WD40 and, **f,** ROC-CORB used in model building with corresponding FSC plots and maps colored by local resolution.

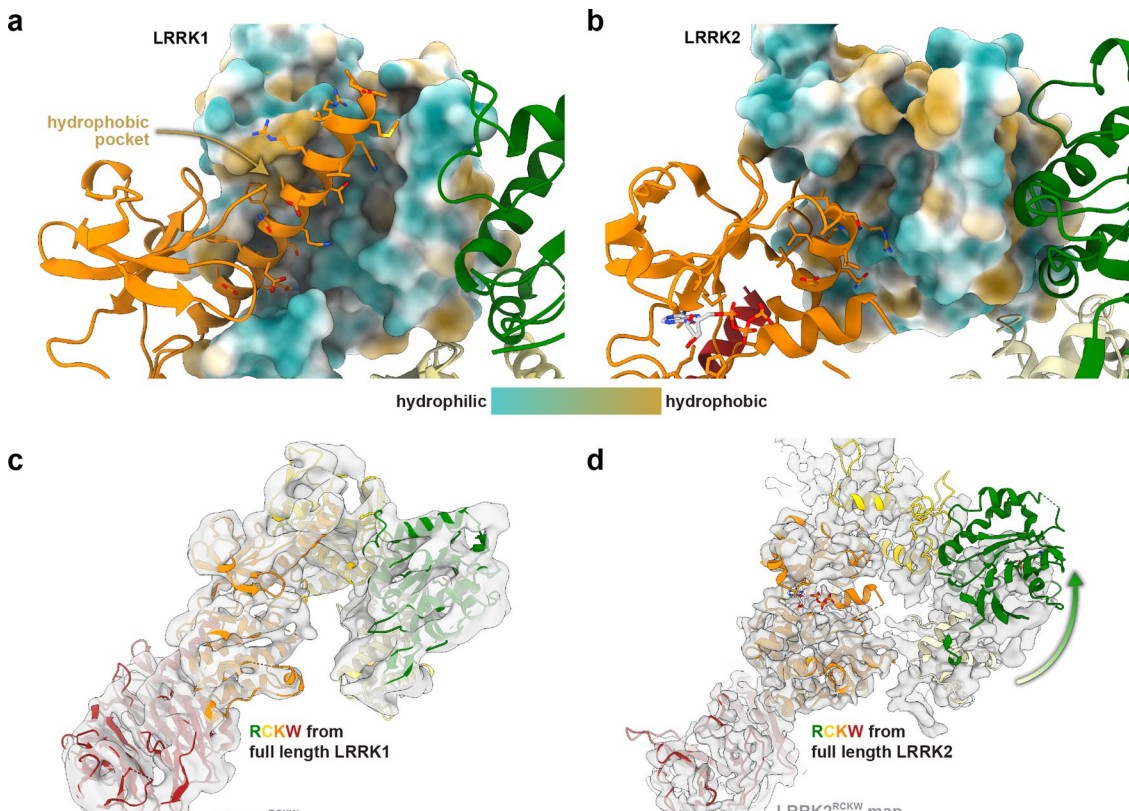

**Extended Data Fig. 2 | Interface between LRRK1's C-alpha helix and the COR-B domain. a**, The N-lobe of LRRK1's kinase domain is shown in ribbon representation, while the COR-B domain is shown in surface representation colored by its hydrophilicity/hydrophobicity. A hydrophobic pocket in the COR-B domain is indicated, and the side chains for residues in the C-alpha helix are shown. **b**, Equivalent view for LRRK2. **c, d**, The RCKW portion of full-length LRRK1 (c) or LRRK2 (d) were docked into cryo-EM maps of LRRK1$^{RCKW}$ (c) or LRRK2$^{RCKW}$ (d). The arrow in (d) indicates that the ROC domain from full-length LRRK2 is rotated relative to its position in the cryo-EM map of LRRK2$^{RCKW}$.

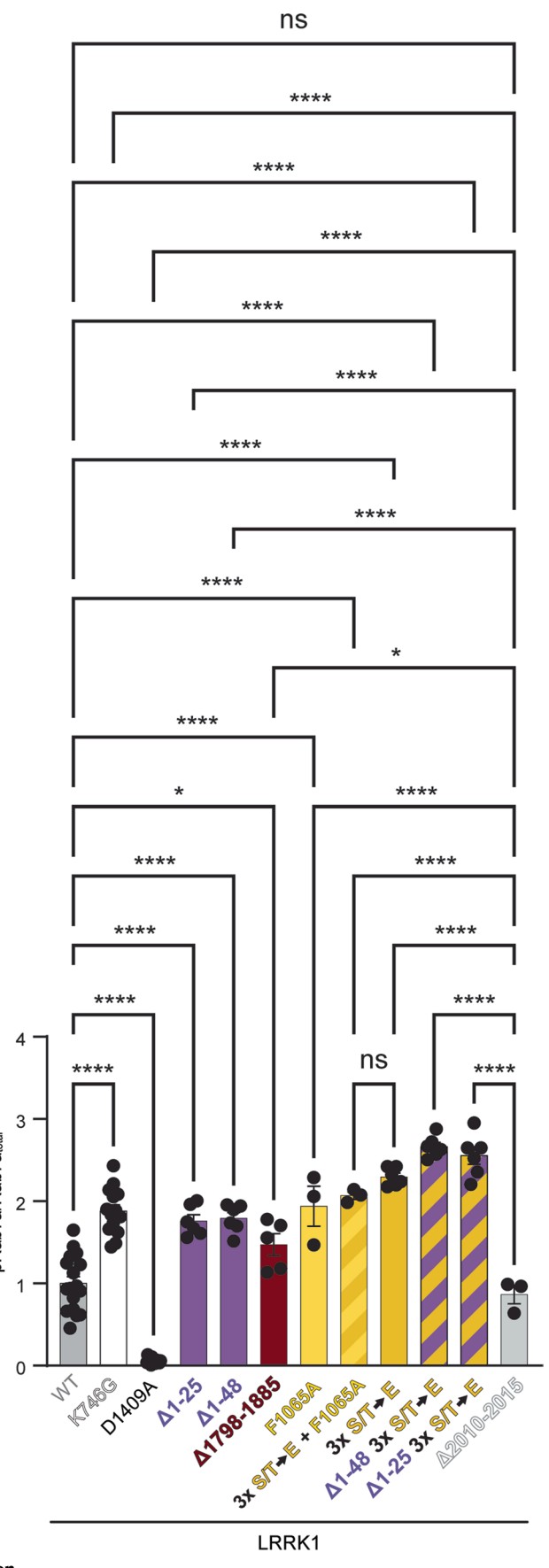

**Extended Data Fig. 3 | See next page for caption.**

**Extended Data Fig. 3 | Rab7a phosphorylation in cells for all constructs analyzed in this work.** Rab7a phosphorylation in 293T cells expressing GFP-Rab7a and one of the following constructs: full-length WT LRRK1 n = 17; LRRK1(K746G) n = 14, which is known to increase Rab7 phosphorylation in cells; LRRK1(D1409A) n = 14, which is known to be kinase inactive; LRRK1(Δ1-25) n = 6; LRRK1(Δ1-48) n = 6; LRRK1(Δ1798-1885) n = 5; LRRK1(F1065A) n = 3; LRRK1(S1064E/F1065A/S1074E/T1075E) n = 3; LRRK1(S1064E/S1074E/T1075E) n = 6; LRRK1(Δ1-25) (S1064E/S1074E/T1075E) n = 6; LRRK1(Δ1-48) (S1064E/

S1074E/T1075E) n = 6; or LRRK1(Δ2010-2015) n = 3. 293T cells were transiently transfected with the indicated plasmids encoding FLAG-LRRK1 (wild type or mutant) and GFP-Rab7. Thirty-six hours post-transfection the cells were lysed, immunoblotted for phosphor-Rab7 (pS72), total GFP-Rab7, and total LRRK1. The mean ± s.e.m. is shown and statistical significance was determined by one way ANOVA with Tukey's multiple comparisons test (*p<0.05, **p<0.01, ***p<0.001, ****p<0.0001, ns = not significant). Individual data points represent separate populations of cells obtained across at least three independent experiments.

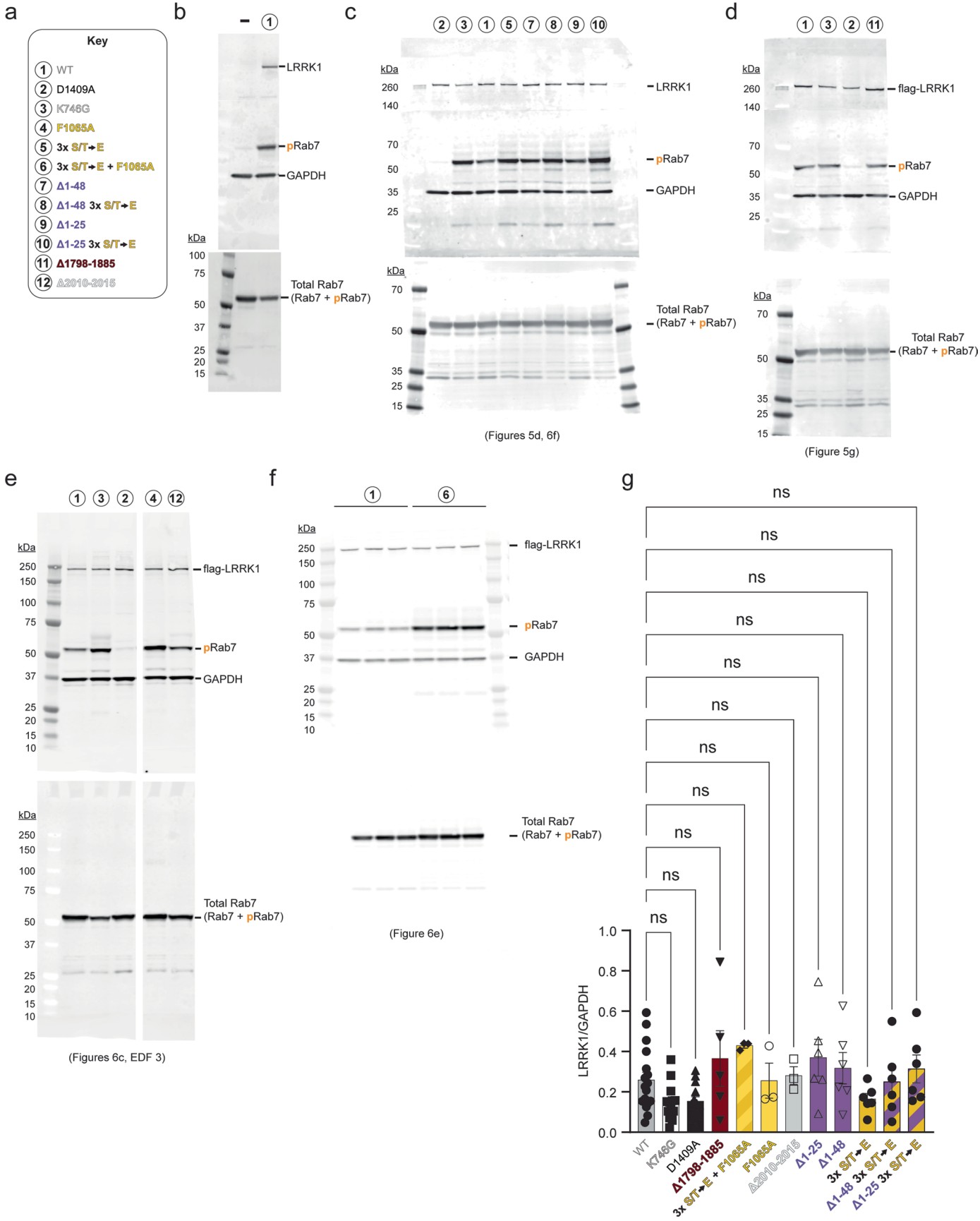

**Extended Data Fig. 4 | See next page for caption.**

**Extended Data Fig. 4 | Expression levels of LRRK1 constructs in 293T cells.** **a**, Key to the different LRRK1 constructs tested in this work and for which data is presented in this figure. **b**–**f**, Representative Western blots of 293T expressing different LRRK1 constructs. The blots were probed with antibodies against LRRK1, GAPDH, phosphorylated Rab7 (pRab7), and total Rab7. No LRRK1 was detected in 293T cells in the absence of overexpression (b). **g**, Quantitation of the overexpression of the different LRRK1 constructs used in this work, normalized relative to WT LRRK1. Expression levels were determined from 3 or more biological replicates. The blots shown in this figure are representative of the data used. Statistics are shown for the same pairwise combinations for which statistics are reported for Rab7 phosphorylation in Figs. 5, 6 and Extended Data Fig. 3. We have indicated below each set of blots the figure(s) where the Rab7 phosphorylation data is shown. Error bars are SEM.

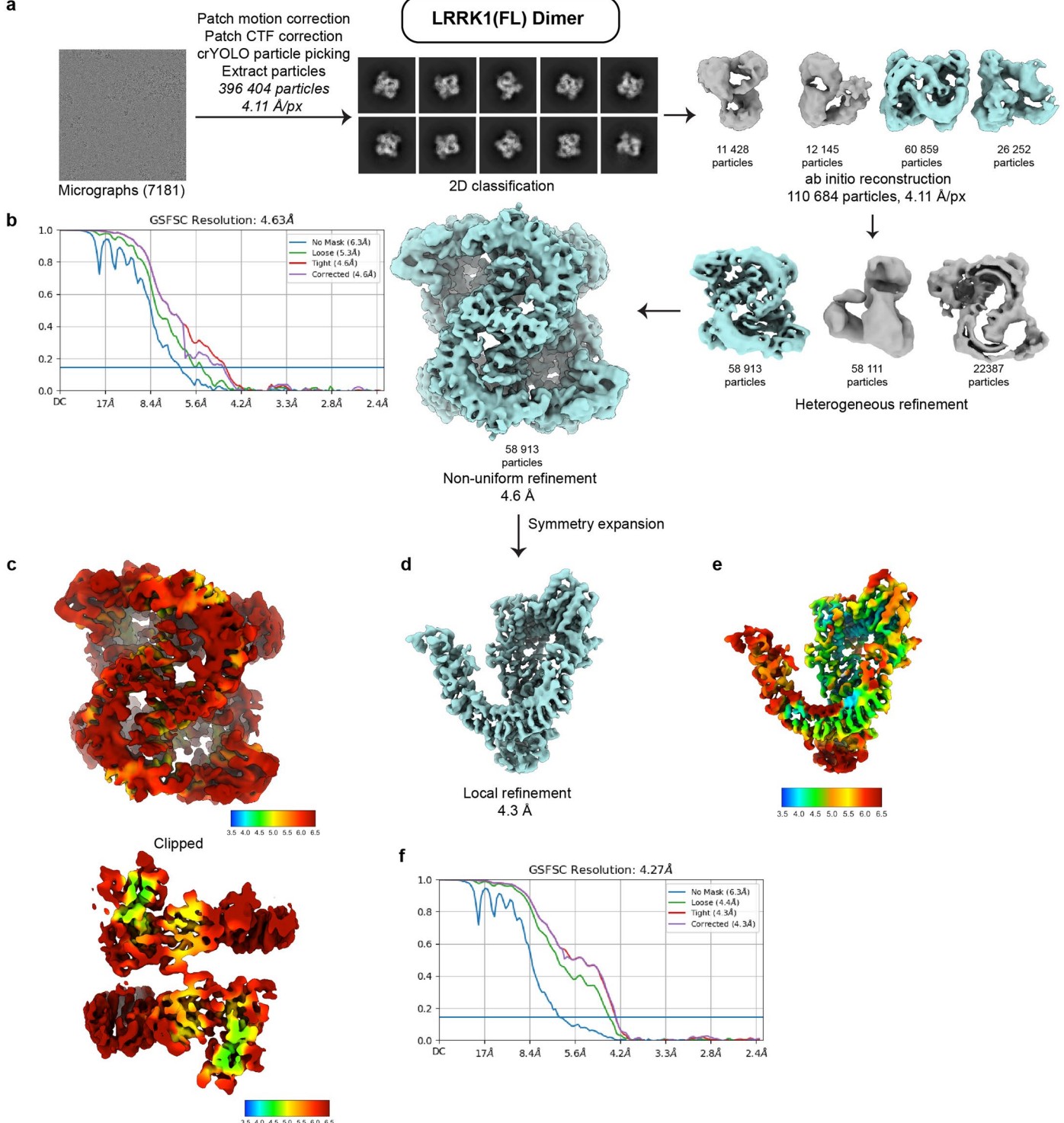

**Extended Data Fig. 5 | Cryo-EM workflow for dimeric LRRK1. a**, Data processing and reconstruction of dimeric LRRK1. **b**, FSC plot for dimeric LRRK1. **c**, Dimer map colored by local resolution (top), with clipped surface (bottom) showing higher resolution in the interior of the map. **d**, Particles were symmetry expanded and used in local refinement to obtain a 4.3Å reconstruction of a LRRK1 monomer. **e**, Symmetry expanded map colored by local resolution. **f**, FSC plot for the symmetry expansion map.

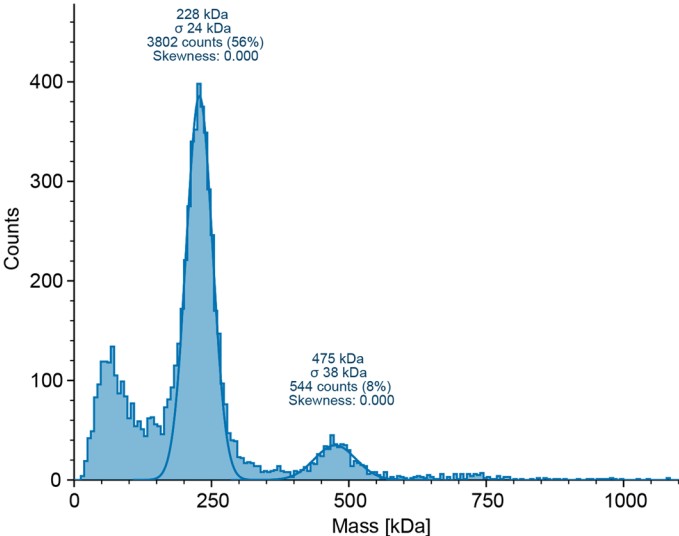

**Extended Data Fig. 6 | Mass photometry of wild-type LRRK1 (full-length).** Wild-type LRRK1 (full-length) was analyzed by mass photometry. Gaussians were fitted to the data to estimate the amount of monomer and dimer in the sample.

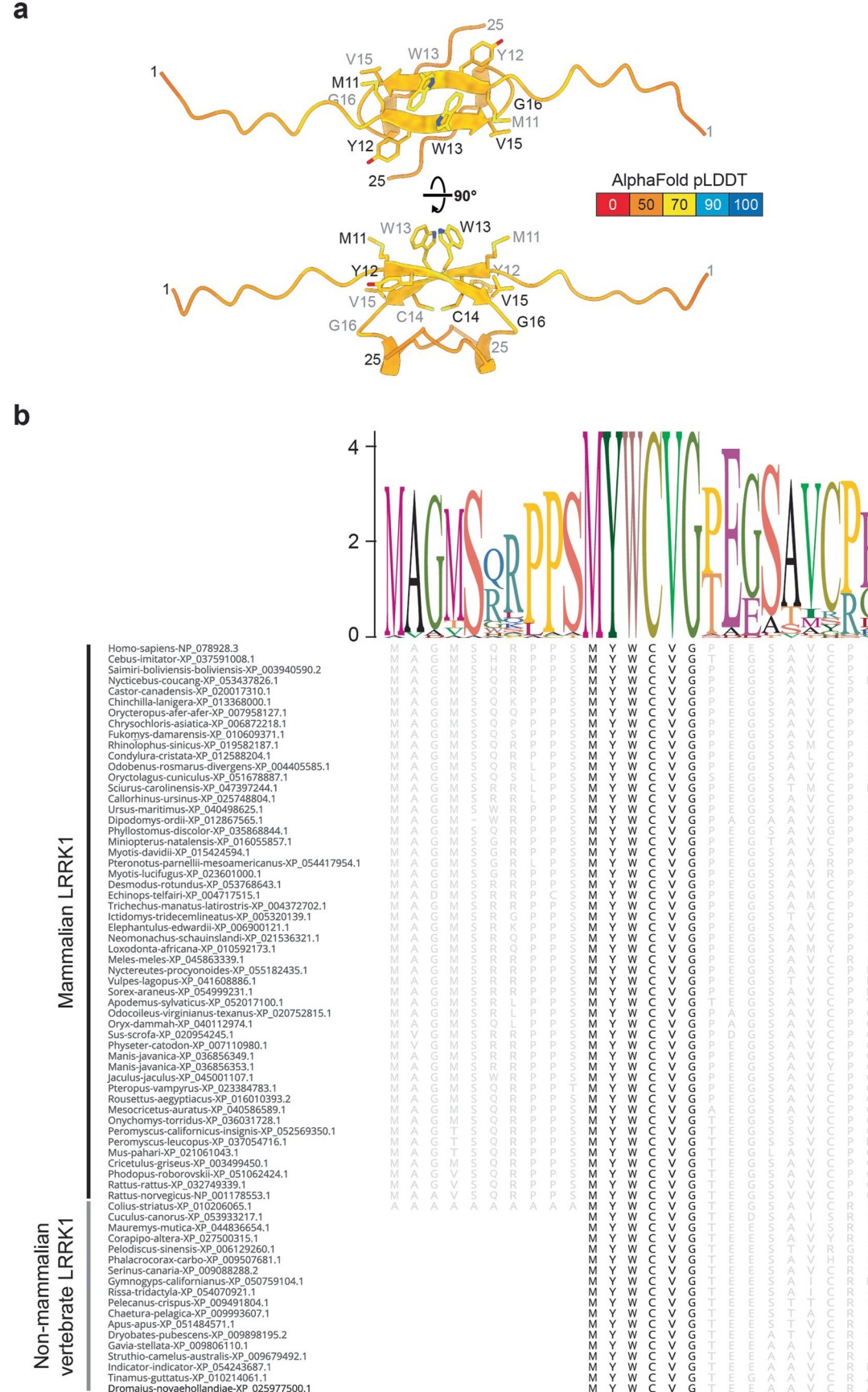

**Extended Data Fig. 7 | See next page for caption.**

**Extended Data Fig. 7 | AlphaFold model of LRRK1's N-terminal dimerization interface. a**, AlphaFold multimer was used to generate a model using two copies of the first 25 residues in LRRK1. The top ranked model is shown in two orientations. The side chains of the residues in the central β-sheet, which are also the most highly conserved, are shown. The structure is colored according to AlphaFold's per-residue confidence value (pLDDT), and the color key is shown on the right. **b**, Sequence alignment of the first 40 residues of vertebrate LRRK1s. The highly conserved MYWCVG motif, corresponding to the β-sheet in (a) is highlighted.

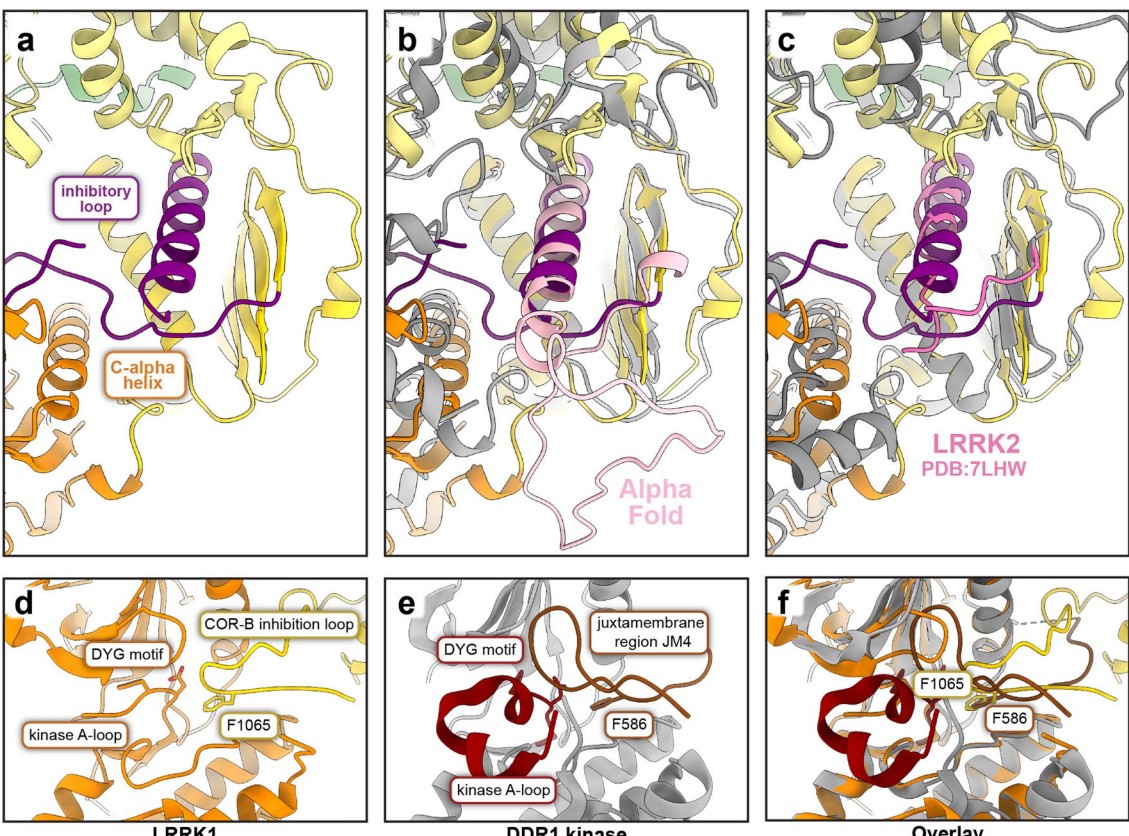

**Extended Data Fig. 8 | LRRK1's COR-B autoinhibitory loop. a–c**, Close ups of the kinase-COR-B region of LRRK1. a, The COR-B autoinhibitory loop is shown in dark purple. b, Same as (a), with the AlphaFold model of LRRK1 shown in light purple. The portion corresponding to the autoinhibitory loop identified in our structure was modeled as disordered in the AlphaFold structure. c, The model for full-length LRRK2 (PDB: 7LHW), in medium purple, is overlayed on the structure of LRRK1 shown in panel (a); the long COR-B loop present in LRRK1 is absent in LRRK2. **d–f**, DDR1 kinase uses an autoinhibitory mechanism analogous to that of LRRK1. d, View of the active site of LRRK1's kinase, similar to Fig. 6d. e, Active site of DDR1 kinase (PDB: 6Y23). f, Overlay of LRRK1 and DDR1. Note that F1065 in LRRK1 and F586 in DDR1 both occupy the same back pocket in the kinase's active site.

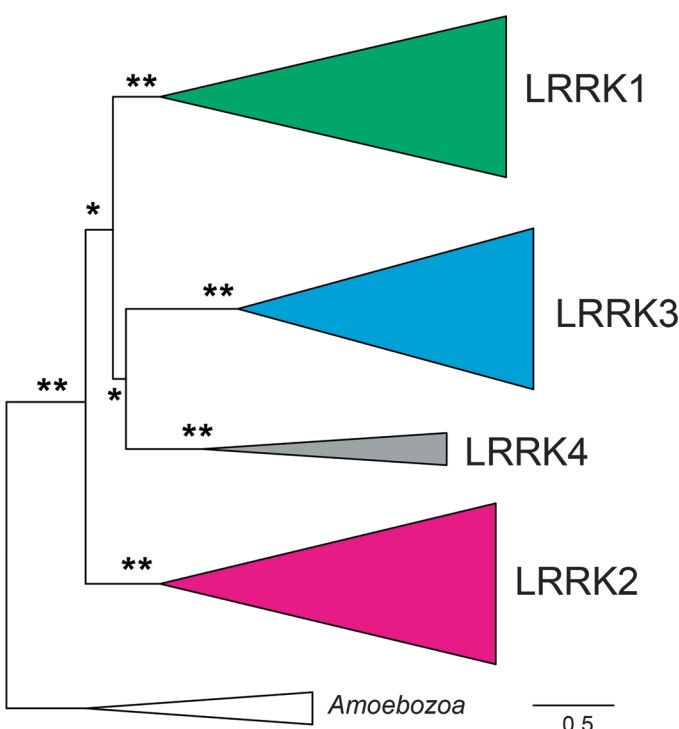

**Extended Data Fig. 9 | Maximum likelihood phylogenetic tree of LRRK protein homologs based only on ROC, COR-A, and kinase domain regions.** An alignment of full length LRRK protein homologs (protein accessions and alignment in Supplementary Datasets 1 And 2, respectively) was generated and then alignment regions corresponding to the ROC-COR-A domains (human LRRK1 residues 632-995) and kinase domain region (human LRRK1 residues 1229-1534) were extracted and concatenated. This extracted alignment, which only contains regions that are shared across all LRRK protein homologs, was used as input to generate a maximum likelihood phylogenetic tree of LRRK protein homologs (complete tree in Supplementary Dataset 3). Shading and rooting are the same as in Fig. 7. Asterisks indicate bootstrap branch support (* >75% support, ** 100% support). Support for major LRRK protein clades is similar to the phylogenetic tree shown in Fig. 7, which was generated from the full-length protein alignment. Human LRRK1 and LRRK2 are found in distinct clades that contain proteins from vertebrates, echinoderms (for example, starfish), spiralians (for example, mollusks), and cnidarians (for example, corals). Arthropod and nematode LRRK proteins are found only in a third clade (labeled LRRK3), which is distinct from vertebrate LRRK1 and LRRK2, and that also contains proteins from echinoderms, spiralians, and cnidarians. Finally, a fourth clade (labeled LRRK4) contains only proteins from cnidarians.

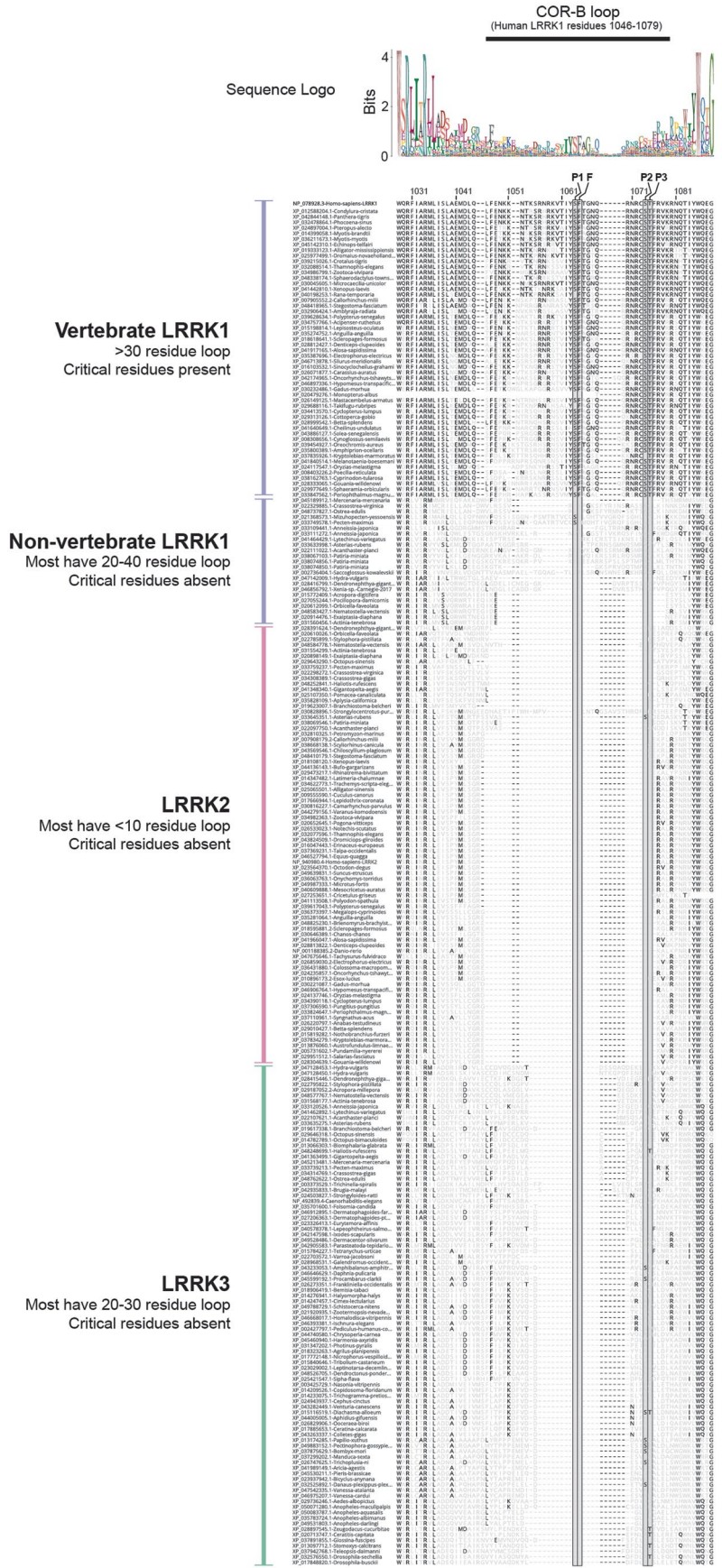

**Extended Data Fig. 10 | See next page for caption.**

**Extended Data Fig. 10 | COR-B loop alignments in LRRK1, LRRK2, and LRRK3 proteins.** Sequence alignment of the region of metazoan LRRK proteins that corresponds to the LRRK1 COR-B inhibitory loop (residues 1046-1079 in human LRRK1) and surrounding residues. The consensus sequence logo is shown at the top, with the alignment below. The Phe that occupies the back pocket of the kinase in the autoinhibited conformation (F1065 in human LRRK1) and the 3 sites of PKC phosphorylation (S1064, S1074, and T1075 in human LRRK1), are indicated above and highlighted in the alignment. Major species clades are shown at the left next to individual sequence accession numbers and species names. The human LRRK1 sequence is shown at the top of the alignment with residue numbers indicated. For other sequences in the alignment, residues in black are identical to human LRRK1. Residues in grey do not match the amino acid of human LRRK1, and dashes indicate there is no amino acid in this alignment position. LRRK1 proteins have a COR-B loop of >27 residues and retain the Phe and phosphorylation sites present in human LRRK1. Other LRRK proteins either have a 'short' COR-B loop (for example, LRRK2) or lack the Phe and phosphorylation sites. LRRK3s, including those from arthropods and nematodes, have an intermediate length loop, ranging from 20–30 residues, whereas most LRRK4s have a much longer insert (200 or more residues) in the COR-B domain (Fig. 7f). The functional consequences of these differences in COR-B remain to be determined.

# Reporting Summary

## Statistics

For all statistical analyses, confirm that the following items are present in the figure legend, table legend, main text, or Methods section.

| n/a | Confirmed | |
|---|---|---|
| ☐ | ☒ | The exact sample size (*n*) for each experimental group/condition, given as a discrete number and unit of measurement |
| ☐ | ☒ | A statement on whether measurements were taken from distinct samples or whether the same sample was measured repeatedly |
| ☐ | ☒ | The statistical test(s) used AND whether they are one- or two-sided *Only common tests should be described solely by name; describe more complex techniques in the Methods section.* |
| ☒ | ☐ | A description of all covariates tested |
| ☐ | ☒ | A description of any assumptions or corrections, such as tests of normality and adjustment for multiple comparisons |
| ☐ | ☒ | A full description of the statistical parameters including central tendency (e.g. means) or other basic estimates (e.g. regression coefficient) AND variation (e.g. standard deviation) or associated estimates of uncertainty (e.g. confidence intervals) |
| ☐ | ☒ | For null hypothesis testing, the test statistic (e.g. *F*, *t*, *r*) with confidence intervals, effect sizes, degrees of freedom and *P* value noted *Give P values as exact values whenever suitable.* |
| ☒ | ☐ | For Bayesian analysis, information on the choice of priors and Markov chain Monte Carlo settings |
| ☒ | ☐ | For hierarchical and complex designs, identification of the appropriate level for tests and full reporting of outcomes |
| ☒ | ☐ | Estimates of effect sizes (e.g. Cohen's *d*, Pearson's *r*), indicating how they were calculated |

*Our web collection on statistics for biologists contains articles on many of the points above.*

## Software and code

Policy information about availability of computer code

Data collection: Leginon was used for all automated EM data collection. For light microscopy experiments, data was collected on commercially available Nikon Elements Software. For Western blots, data was collected using Image Studio v5.2 (Li-COR)

Data analysis: CryoSPARC Live was used to align movie frames. Particle picking was done with crYOLO. Subsequent data processing was done using Relion 3.0 and cryoSPARC.

For Western blot, data was quantified using ImageStudio v5.2. and ImageJ

Phylogenetic analysis was done using BLASTp for searching and Clustal Omega for alignment. IQ-TREE was used to generate maximum likelihood phylogenetic trees. Consensus logos and alignment visualization was done using Geneious Prime.

For manuscripts utilizing custom algorithms or software that are central to the research but not yet described in published literature, software must be made available to editors and reviewers. We strongly encourage code deposition in a community repository (e.g. GitHub). See the Nature Portfolio guidelines for submitting code & software for further information.

## Data

Policy information about availability of data

All manuscripts must include a data availability statement. This statement should provide the following information, where applicable:

- Accession codes, unique identifiers, or web links for publicly available datasets
- A description of any restrictions on data availability
- For clinical datasets or third party data, please ensure that the statement adheres to our policy

All raw data that went into the biochemical and cell biological analyses were deposited in a spreadsheet with the manuscript.

# Field-specific reporting

Please select the one below that is the best fit for your research. If you are not sure, read the appropriate sections before making your selection.

☒ Life sciences ☐ Behavioural & social sciences ☐ Ecological, evolutionary & environmental sciences

For a reference copy of the document with all sections, see nature.com/documents/nr-reporting-summary-flat.pdf

# Life sciences study design

All studies must disclose on these points even when the disclosure is negative.

| | |
|---|---|
| Sample size | For all experiments, we determined sample size following established conventions in the field. |
| Data exclusions | No data were excluded. |
| Replication | Rab7a phosphorylation assays by Western blot in Figures 5, 6, S3, & S4 were performed with between three and eight biological replicates. Replicate numbers for each figure are indicated in the raw data spreadsheet deposited with the manuscript. All attempts at replication were successful. |
| Randomization | This is not relevant. We have no data involving organisms or subjects that would require randomization. |
| Blinding | None |

# Reporting for specific materials, systems and methods

We require information from authors about some types of materials, experimental systems and methods used in many studies. Here, indicate whether each material, system or method listed is relevant to your study. If you are not sure if a list item applies to your research, read the appropriate section before selecting a response.

### Materials & experimental systems

| n/a | Involved in the study |
|---|---|
| ☐ | ☒ Antibodies |
| ☐ | ☒ Eukaryotic cell lines |
| ☒ | ☐ Palaeontology and archaeology |
| ☒ | ☐ Animals and other organisms |
| ☒ | ☐ Human research participants |
| ☒ | ☐ Clinical data |
| ☒ | ☐ Dual use research of concern |

### Methods

| n/a | Involved in the study |
|---|---|
| ☒ | ☐ ChIP-seq |
| ☒ | ☐ Flow cytometry |
| ☒ | ☐ MRI-based neuroimaging |

## Antibodies

| | |
|---|---|
| Antibodies used | Rabbit anti-LRRK1 antibody (ab228666), rabbit anti-GAPDH (Cell Signaling Technology, 14C10, Lot 14-2118S); mouse anti-GFP (Santa Cruz, clone: B-2, Cat: sc-9996, Lot: C1518); IRDye 800CW goat anti-rabbit (LiCOR, P/N:; 926-32211, Lot: C90229-05); IRDye 680RD goat anti-mouse (LiCOR, P/N: 926-68070, Lot: C90219-05) |
| Validation | All antibodies used are well-validated and highly specific commercially available antibodies. For LiCOR quantification, linear range was determined for each antibody. |

# Eukaryotic cell lines

Policy information about [cell lines](cell lines)

| | |
|---|---|
| Cell line source(s) | HEK293T used were from ATCC (CRL-3216) |
| Authentication | ATCC authenticated |
| Mycoplasma contamination | New cell lines received by our lab are tested for mycoplasma before expanding and freezing. After thawing, each cell line is tested again. Every three months, all cells growing in the lab are tested for mycoplasma as well. The cells used in our experiments were last tested on 06/23/21 and did not contain contamination. |
| Commonly misidentified lines (See ICLAC register) | No commonly misidentified cell lines were used. |

