## [Peer Review File · Nature Structural & Molecular Biology]

Peer Review Information

Manuscript Title: Structure of LRRK1 and mechanisms of autoinhibition and activation

Corresponding author name(s): Andres Leschziner, Samara Reck-Peterson

Reviewer Comments & Decisions:

Decision Letter, initial version:

Message: 28th Apr 2023

Dear Dr. Leschziner,

Thank you again for submitting your manuscript "Structure of LRRK1 and mechanisms of autoinhibition and activation". I apologize for the delay in responding, which resulted from the difficulty in obtaining suitable referee reports. Nevertheless, we now have comments (below) from the 3 reviewers who evaluated your paper. In light of those reports, we remain interested in your study and would like to see your response to the comments of the referees, in the form of a revised manuscript.

You will see that while all reviewers find the results timely and interesting, they bring up concerns which we will expect to be addressed in a revision. Specifically, reviewer #2 comments on the lack insights into pathogenesis in the manuscript, echoing our initial concerns. We will encourage you to discuss the aspect further and explore it experimentally. In line with reviewer's #3 comments, we agree that further investigation into the role of dimerization of LRRK1 in regulating its function in cellular context will strengthen the manuscript.

Please be sure to address/respond to all concerns of the referees in full in a point-by-point response and highlight all changes in the revised manuscript text file. If you have comments that are intended for editors only, please include those in a separate cover letter.

We expect to see your revised manuscript within 6 weeks. If you cannot send it within this

time, please contact us to discuss an extension; we would still consider your revision, provided that no similar work has been accepted for publication at NSMB or published elsewhere.

Reporting Summary:

Data availability: this journal strongly supports public availability of data. All data used in accepted papers should be available via a public data repository, or alternatively, as Supplementary Information. If data can only be shared on request, please explain why in

your Data Availability Statement, and also in the correspondence with your editor. Please note that for some data types, deposition in a public repository is mandatory - more information on our data deposition policies and available repositories can be found below: <https://www.nature.com/nature-research/editorial-policies/reporting-standards#availability-of-data>

[redacted]

Sincerely,
Kat

Katarzyna Ciazynska
(she/her)
Associate Editor
Nature Structural & Molecular Biology
<https://orcid.org/0000-0002-9899-2428>

Referee expertise:

Referee #1: Parkinson's disease, molecular biology

Referee #2: LRRK1, signal transduction

Referee #3: LRRK proteins, cryo-EM

Reviewers' Comments:

Reviewer #1:

Remarks to the Author:

This manuscript presents the structures of LRRK1 monomers and dimers in relation to the structure of LRRK2 that is implicated in Parkinson's disease. Comparison of the structures reveals important differences in kinase regulation and the associated evolutionary data highlights the fact that more distant LRRK2 relatives are closer to LRRK1 than LRRK2. The data are outstanding; the manuscript is well written and the story will be of broad interest to readers of NSMB. Thus the story should be accepted for publication in its current form, without delay.

Reviewer #2:

Remarks to the Author:

In this paper, the authors analyzed the structure of full-length LRRK1, which lags behind LRRK2, and attempted to clarify why only LRRK2 is involved in Parkinson's Disease, even though LRRK1 and LRRK2 have similar domain architectures and related cellular functions. Using cryo-EM analysis, they revealed two autoinhibitory mechanisms in LRRK1: (1) autoinhibition by the COR-B loop and (2) dimer-dependent steric autoinhibition of the kinase by the ANK domain. These findings are very interesting because they are regulatory mechanisms specific to LRRK1. Therefore, this paper deserves to be ACCEPTED. However, a weakness of this study is that these structural findings do not provide new insights into the pathogenesis of diseases involving LRRK1 or LRRK2. The same is true for the structural motif analysis of LRRK family proteins in the second half of this study. Because of these weaknesses, the study may not be of sufficient interest to researchers in a wide range of biological research fields. It may be better to submit to a journal specifically devoted to structural analysis.

Reviewer #3:

Remarks to the Author:

The paper describes the structure-function characterization of human LRRK1 and its comparison with LRRK2. The authors describe striking differences, in spite of similar domain organization of target proteins, that have important repercussions for the functioning and regulation of these proteins in vivo. The data is beautifully presented and the paper is a pleasure to read. I believe it is a significant contribution to the field that will

be broadly appreciated by your readership. I just have some comments that the authors may want to address before publication. They are listed below in an order that reflects the manuscript structure, not the significance of the comments themselves:

- The authors added ATP and GTP. Given that the GTP was hydrolyzed, would adding a non-hydrolysable analog make a difference in the conformation of the monomer or dimer structures that may be of any relevance? I do not know what the function of the GTPase domain is.
- N-terminal residues and dimerization: Given that the authors limited the requirement for dimerization (based on de-inhibition of the kinase activity in cells) to the first 25 a.a., can the authors not propose a model of the unassigned density at the interface? In fact, given that their original 1-19 truncation gives an intermediate result (with a small number of dimers), the region between 19-25 is likely to be part of the density. How conserved are residues 1-25? What is the chemical nature of the other side of the interface (could they infer what kind of residues should be making the contacts)? Have they tried to see the fold of these residues using alphafold multimer? It obviously involves dimerization of the protein. If alpha fold can predict the dimerization, it may also predict the folding of those a.a.s.
- Have the authors tested that the LRRK1(Δ 1798-1885) construct actually results in less dimers in vitro?
- Does the blurry density proposed to correspond to that region change when no symmetry is applied? If symmetry expansion is applied?
- Why was symmetry expansion and focused refinement needed to provide an improved density for the COR-B loop (residues 1048-1082)? Is this engaged in a non-symmetrical manner? Is it occupying the site by the kinase in only some molecules?
- A question that arises given the many ways LRRK1 is autoinhibited is how it can have some significant activity in cells. Phosphorylation of the COR-B loop is a great solution for one of the inhibition mechanisms, but something else is required to explain the release of the dimerization. Also, the fact that mutants that abolish each of them are additive, but that effect of the dimer mutants in the background on the loop phospho-mutants is so small may point to some coupling. In fact, the authors indicate in the discussion that their structural data suggest that disruption of the dimer should precede phosphorylation of the COR-B residues that lead to activation. Please see following comment on monomer-dimer equilibrium in the cell, which will apply to the comment I just made.
- In the discussion the authors remark how eliminating either of the dimerization interfaces resulted in only a modest increased Rab7a phosphorylation in cells and suggested that other interfaces may still play a major role in stabilizing the LRRK1 dimer. Given that there is another component to the inhibition and that the interfering with both shows the larger effect for the Cor-B loop phosphor-mutant (see comment above), it may be that eliminating dimerization is not as relevant for activity altogether. This could be because dimerization is already regulated in the cell in a way that is lost in the simplified in vitro system (perhaps by the proposed phosphorylation of the interfaces -obviously, the authors could have generated mutants to test this hypothesis), or because the concentration in cells of the protein leads to an equilibrium between monomers and

dimers not seen at higher, in vitro concentrations. The authors could get an idea of the K_D of dimerization and compare it with cellular concentrations of the protein.

Author Rebuttal to Initial comments

A point-by-point response to reviewers follows. We have highlighted new additions to the revised manuscript in the response to review, and those are indicated in the manuscript as well as tracked changes.

Reviewer #1:

Remarks to the Author:

This manuscript presents the structures of LRRK1 monomers and dimers in relation to the structure of LRRK2 that is implicated in Parkinson's disease. Comparison of the structures reveals important differences in kinase regulation and the associated evolutionary data highlights the fact that more distant LRRK2 relatives are closer to LRRK1 than LRRK2. The data are outstanding; the manuscript is well written and the story will be of broad interest to readers of NSMB. Thus the story should be accepted for publication in its current form, without delay.

We thank the reviewer for such positive feedback.

Reviewer #2:

Remarks to the Author:

In this paper, the authors analyzed the structure of full-length LRRK1, which lags behind LRRK2, and attempted to clarify why only LRRK2 is involved in Parkinson's Disease, even though LRRK1 and LRRK2 have similar domain architectures and related cellular functions. Using cryo-EM analysis, they revealed two autoinhibitory mechanisms in LRRK1: (1) autoinhibition by the COR-B loop and (2) dimer-dependent steric autoinhibition of the kinase by the ANK domain. These findings are very interesting because they are regulatory mechanisms specific to LRRK1. Therefore, this paper deserves to be ACCEPTED. However, a weakness of this study is that these structural findings do not provide new insights into the pathogenesis of diseases involving LRRK1 or LRRK2. The same is true for the structural motif analysis of LRRK family proteins in the second half of this study. Because of these weaknesses, the study may not be of sufficient interest to researchers in a wide range of biological research fields. It may be better to submit to a journal specifically devoted to structural analysis.

We thank the reviewer for the positive feedback.

We would like to clarify that our goal was to enable, rather than to directly provide, major insights into the molecular basis of the pathogenesis driven by LRRK1 and LRRK2. Understanding the diseases driven by LRRK1 and LRRK2 would be well beyond the scope of a single manuscript, and we apologize if we gave the impression that this was our goal. Instead, we set out to understand what structural and mechanistic features are common to LRRK1 and LRRK2 and, more importantly, which ones are different. We believe that the impact of our work lies in that it provides a blueprint for researchers interested in the molecular and cellular basis of Parkinson's Disease or LRRK1-linked bone diseases; we have identified LRRK1- and LRRK2-specific features that can be targeted in functional studies. We envisioned our work as foundational, providing two different fields with the tools needed to eventually understand the molecular basis of pathogenesis. We added a sentence to the Discussion to clarify this as well.

LRRK1. Since LRRK1-associated bone diseases are loss-of-function, understanding pathogenesis will require dissecting LRRK1's normal cellular functions. Our work suggests that LRRK1 is tightly regulated (potentially more so than LRRK2, based on the data so far), which is consistent with the fact that no gain-of-function disease mutations have been identified so far. The structural and functional data we presented revealed how LRRK1 is regulated, and our evolutionary analysis has pointed out features that

are LRRK1-specific. Our work provides researchers the tools needed to engineer activating mutations in LRRK1, which will be critical to study its function.

LRRK2. Our structures of LRRK1 have shown that, despite their apparent similarity in terms of domain architecture, LRRK1 and LRRK2 have different structures and organizations at the monomer and dimer levels, as well as different mechanisms of regulation. Our structure-guided evolutionary analysis allowed us to show that the basic patches in the ROC domain as a LRRK2-specific feature. We expect that many groups interested in LRRK2 and its connection to Parkinson's Disease will work on establishing the importance of these motifs in LRRK2's normal and pathogenic roles.

Reviewer #3:

Remarks to the Author:

The paper describes the structure-function characterization of human LRRK1 and its comparison with LRRK2. The authors describe striking differences, in spite of similar domain organization of target proteins, that have important repercussions for the functioning and regulation of these proteins in vivo. The data is beautifully presented and the paper is a pleasure to read. I believe it is a significant contribution to the field that will be broadly appreciated by your readership. I just have some comments that the authors may want to address before publication. They are listed below in an order that reflects the manuscript structure, not the significance of the comments themselves:

We thank the reviewer for such positive feedback.

- The authors added ATP and GTP. Given that the GTP was hydrolyzed, would adding a non-hydrolysable analog make a difference in the conformation of the monomer or dimer structures that may be of any relevance? I do not know what the function of the GTPase domain is.

Although the role of the GTPase domain is of much interest to many people working on LRRK1 and LRRK2, including us, and likely important in their regulation, it remains largely a mystery. Even more so for LRRK1, since much less is known about it than about LRRK2. There is data suggesting that the nucleotide state of the GTPase regulates the kinase, but this is not a settled matter and there is no structural data yet showing how this regulation would take place.

We purify LRRK1 in the presence of GDP but have tested adding other guanine nucleotides when preparing cryo-EM grids. We have imaged LRRK1 in the presence of GTP (the structures reported in our manuscript), as well as with GTP γ S and GDP. We did not see any obvious difference among the 3 samples at the level of 2D class averages or monomer/dimer distribution. We proceeded with the GTP sample simply because those grids turned out to be the best and led to the highest resolution structures.

- N-terminal residues and dimerization: Given that the authors limited the requirement for dimerization (based on de-inhibition of the kinase activity in cells) to the first 25 a.a., can the authors not propose a model of the unassigned density at the interface? In fact, given that their original 1-19 truncation gives an intermediate result (with a small number of dimers), the region between 19-25 is likely to be part of the density. How conserved are residues 1-25? What is the chemical nature of the other side of the interface (could they infer what kind of residues should be making the contacts)? Have they tried to see the fold of these residues using alphaFold multimer? It obviously involves dimerization of the protein. If alpha fold can predict the dimerization, it may also predict the folding of those a.a.s.

We thank the reviewer for these questions.

The reason why we did not use AlphaFold initially to try to build a model for the unassigned density is that it is our policy not to build models unless we are confident about the assignment. Sometimes we only model secondary structure elements as poly-Ala chains if we don't have the resolution to see side chains. If we're not even confident about secondary structure, we don't build that part of the model. In this case, the N-terminal interface of the dimer is one of the worst parts of the map, and AlphaFold predicted the first 1-48 residues to be disordered. It didn't occur to us at the time to try AlphaFold multimer with the N-terminal part of LRRK1, a very good suggestion from the reviewer.

We have now generated an AlphaFold model for the dimer interface involving these residues. Although the per-residue confidence values are low (in the 34-56% range), the region with the highest values happens to form a β -strand, which comes together with its symmetry mate to form a central β -sheet at the modeled interface. Even more interestingly, this motif (MYWCVG) is universally conserved among vertebrate LRRK1s. While the AlphaFold confidence values would make us skeptical on their own, the structural model in combination with the sequence alignment make a strong case for the importance of this region in dimerization.

We include a **new supplementary figure (Extended Data Figure 7)** in the revised manuscript showing the sequence conservation and the AlphaFold model. These results are **mentioned in the text as well**.

- Have the authors tested that the LRRK1(Δ 1798-1885) construct actually results in less dimers in vitro?

We have not tested the effect of deleting this loop (or residues 1-25) on dimerization in vitro. This is largely because our experiments were driven by our hypothesis that these different dimerization interfaces were involved in stabilizing the autoinhibited dimer; thus, we chose to test Rab7a phosphorylation, the product of activation.

- Does the blurry density proposed to correspond to that region change when no symmetry is applied? If symmetry expansion is applied?

The blurry density remained relatively constant during data processing. We tried several different refinement strategies (with and without symmetry applied) and nothing made it appreciably better or worse.

- Why was symmetry expansion and focused refinement needed to provide an improved density for the COR-B loop (residues 1048-1082)? Is this engaged in a non-symmetrical manner? Is it occupying the site by the kinase in only some molecules?

Although we cannot rule out the possibilities the reviewer brings up, we think the main reason why symmetry expansion helped is that there is some flexibility between the two LRRK1 monomers in the dimer. Symmetry expansion allowed us to align monomers better to each other.

The COR-B loop clearly has some intrinsic flexibility, which will therefore be asymmetric in the context of the dimer. While our data so far does not suggest that the important interactions involving the COR-B

loop, particularly the F1065 “plug” in the kinase active site, are asymmetric, we cannot rule out that asymmetry in the disordered regions of COR-B plays a physiological role in regulation.

- A question that arises given the many ways LRRK1 is autoinhibited is how it can have some significant activity in cells. Phosphorylation of the COR-B loop is a great solution for one of the inhibition mechanisms, but something else is required to explain the release of the dimerization. Also, the fact that mutants that abolish each of them are additive, but that effect of the dimer mutants in the background on the loop phospho-mutants is so small may point to some coupling. In fact, the authors indicate in the discussion that their structural data suggest that disruption of the dimer should precede phosphorylation of the COR-B residues that lead to activation. Please see following comment on monomer-dimer equilibrium in the cell, which will apply to the comment I just made.

We agree with the reviewer that the most interesting (and puzzling) aspect of LRRK1 based on our structural data is how the protein becomes fully activated. While it is true that the overall activation we saw when we combined the N-terminal deletion with the phosphomimetic mutations is modest, we should point out that this construct does not come close to mimicking what we think a fully activated species might look like. First, this construct still contains the 1798-1885 loop, which our data suggest contributes to dimerization. Second, the phosphomimetic S/T → E mutations are only a crude approximation of phosphates and thus likely much weaker in terms of activation.

- In the discussion the authors remark how eliminating either of the dimerization interfaces resulted in only a modest increased Rab7a phosphorylation in cells and suggested that other interfaces may still play a major role in stabilizing the LRRK1 dimer. Given that there is another component to the inhibition and that the interfering with both shows the larger effect for the Cor-B loop phosphor-mutant (see comment above), it may be that eliminating dimerization is not as relevant for activity altogether. This could be because dimerization is already regulated in the cell in a way that is lost in the simplified in vitro system (perhaps by the proposed phosphorylation of the interfaces -obviously, the authors could have generated mutants to test this hypothesis), or because the concentration in cells of the protein leads to an equilibrium between monomers and dimers not seen at higher, in vitro concentrations. The authors could get an idea of the kD of dimerization and compare it with cellular concentrations of the protein.

The reviewer raises several interesting and important points; we have revised our Discussion to clarify some aspects based on the reviewer’s comments.

We engineered constructs to test the roles of several structural features in regulating LRRK1’s kinase activity. We hypothesize that this regulation happens at two different levels: first, by stabilizing the autoinhibited dimer (here, we tested the N-terminus and the 1798-1885 loop), and then by directly blocking the kinase’s active site (where we tested the F1065 “plug”, and the phosphorylation sites S1064/S1074/T1075 in the COR-B loop). As pointed out in our original Discussion, regulation by the COR-B loop appears to be independent of dimerization, as we see the loop, and the F1065 plug, in a similar conformation in both monomer and dimer. Disruption of the dimer would remove the steric autoinhibition (the blocking of one kinase active site by the ANK domain of the other monomer, in trans), but an additional activation step (at a minimum, phosphorylation) would be required to remove the F1065 plug from the kinase’s active site. Therefore, we would expect that disruption of the dimer is necessary, but not sufficient to activate LRRK1.

What we failed to emphasize more clearly in the Discussion is that we did not test the level of Rab7a phosphorylation with a construct where all putative dimer-stabilizing interactions had been disrupted. A major reason for that is that our goal was to test whether certain individual features contributed to dimer formation, and thus autoinhibition, and we focused on the two regions of the map where we could not build a model at all (the N-terminus and the WD40 loop 1798-1885). While the other interfaces involved regions of the map where we could build a model, the resolutions were not high enough to confidently engineer mutations that could test those additional 5 dimer interfaces: ANK_A-ANK_B; LRR_A-ANK_B; LRR_A-Kinase_A-ANK_B; Kinase_A-ANK_B; and Kinase_A:Kinase_B (see Figure 4). Along with the two features we did test, that makes for a total of 7 putative interfaces involved in dimer stabilization. It is possible that one would have to disrupt several of these to fully break up the dimer. Even then, that only sets the stage up for the next level of activation: phosphorylation of the COR-B loop. And this is where the phosphomimetic mutants are only an approximation of what true phosphate groups may be able to achieve in terms of destabilizing the loop and relieving the F1065 plug. We have expanded the “LRRK1 regulation” section of the Discussion to make these points more clearly; we realize that the original version may have downplayed the limited and targeted nature of the testing we performed, thus suggesting that the modest increase in Rab7a phosphorylation in cells might have been unexpected.

We also wanted to clarify that none of our measurements of Rab7a phosphorylation were done strictly “in vitro”. While our measurements of Rab7a phosphorylation in cells are not done under native conditions—the protein is overexpressed and we did not use a cell line where LRRK1 is normally expressed at high levels—some of the physiological regulatory components may be present, though arguably in levels that may not match LRRK1 overexpression. Testing regulation of LRRK1 in a physiological cellular environment beyond the structural elements we identified in our manuscript would require that we introduce mutations in the endogenous copy of LRRK1 in a cell line where the protein is normally expressed at higher levels. This is certainly something we are very interested in, but the scope of this work is beyond what we could tackle during the timeline of a revision.

The reason why we did not test phosphorylation of interfaces as a means of regulating dimerization is the vast number of predicted sites. Here is the output from running LRRK1 through the NetPhos prediction server:

```

%1 ...S...S.Y.....T...T...ST..... # 50
%1 .S..T.....S.....S..... # 100
%1 .....S.....S..... # 150
%1 .....SY.....Y. # 200
%1 ..S.....S...S...S...S...SS. # 250
%1 ..T.....S.....S.....T..S... # 300
%1 .....SS...S.....SS...S..... # 350
%1 .S.....T.....S..... # 400
%1 .S..S.....S..... # 450
%1 .....S.....T..... # 500
%1 T.....T.....T.....S.T...S... # 550
%1 .....S...S..... # 600
%1 .T.....S..... # 650
%1 .S.....T.....T..TT.....S..... # 700
%1 .....T...Y..... # 750
%1 .....T.....S.S.S..T..... # 800
%1 .....S.....S.....S.....S..... # 850
%1 .....S...Y.T.....T.....Y...S.. # 900
%1 ..T.....T.....S.S..... # 950
%1 .....T..T..Y.....Y.....S... # 1000
%1 ..T.....T..T...S.....S..... # 1050
%1 ..T.S.....T..S.T.....ST.....T.Y...Y.S # 1100
%1 .....S.....S.....S.....T. # 1150
%1 ..S..T...Y.....S..S...Y.....T. # 1200
%1 ..S.....S.....S.....S..... # 1250
%1 .S.T.....T..... # 1300
%1 .S.....S.....S.....S..T..... # 1350
%1 ..SS.....T...Y..S.....S..... # 1400
%1 .....S.....S.....T..Y.....SY # 1450
%1 .....S.....S..... # 1500
%1 ..T.....S..S.....S.....Y.. # 1550
%1 .....YT...T.....S..... # 1600
%1 .....T.....T.....T.....SY..... # 1650
%1 .....T...S...S.....S..... # 1700
%1 S...Y.....S.....Y..S.....SS. # 1750
%1 .....S.....S.....T.....S # 1800
%1 .....S...S...S.....S.T.Y.S.SSYSS... # 1850
%1 ...S.SS..SS..SSSS..ST...S...T...S...T... # 1900
%1 ...S..... # 1950
%1 ..T.....T...T.....Y.S # 2000
%1 Y.....T....

```

There are many predicted sites with very high phosphorylation potential. In addition to the problem of selecting sites (and their many permutations), we still have the issue that phosphomimetic mutations do not always recapitulate, even partially, the effect of true phosphorylation. (Even though our LRRK1 phosphomimetics did show some effects, we have made phosphomimetic mutations in LRRK2 that showed no phenotype.) The role of phosphorylation in regulating LRRK1 remains an important question; going forward, we plan to address this by starting with phosphoproteomics to characterize the phosphorylation state of LRRK1 in cells.

Following the reviewer's suggestion, we used mass photometry to measure the K_d for LRRK1 dimerization. The estimated value (0.6 μM) is almost 9X lower than the concentration of LRRK1 (full-length) we used to prepare cryo-EM grids, likely explaining why we mostly saw dimers with this construct. We now include a **new Extended Data Fig. 6** showing the mass photometry data, and **new text**

pointing out that although this K_d is relatively high compared to reported concentrations of LRRK1 in cells; the localization of the protein to membranes would likely increase its local concentrations to levels that could drive dimerization.

Finally, we know little about the potential activation of LRRK1 by substrates. In the case of LRRK2, some recent work has identified 3 different binding sites for Rab's on the N-terminal repeats of the protein (more specifically in the Armadillo repeats that are present in LRRK2 but absent in LRRK1) (PMID 36149401 and doi: <https://doi.org/10.1101/2023.02.17.529028>). These binding sites are involved in the recruitment of LRRK2 to membranes and its activation there. These sites are separate from the Rab substrate-binding site that is yet to be characterized. Whether similar mechanisms exist for LRRK1 (involving other parts of the protein) is not yet known.

Decision Letter, first revision:

Message: Our ref: NSMB-A47389A

2nd Jun 2023

Dear Dr. Leschziner,

Thank you for submitting your revised manuscript "Structure of LRRK1 and mechanisms of autoinhibition and activation" (NSMB-A47389A). I am writing to inform you that we'll be happy in principle to publish it in Nature Structural & Molecular Biology, pending minor revisions to comply with our editorial and formatting guidelines.

Thank you again for your interest in Nature Structural & Molecular Biology. Please do not hesitate to contact me if you have any questions.

Sincerely,
Kat

Katarzyna Ciazynska
(she/her)
Associate Editor
Nature Structural & Molecular Biology
<https://orcid.org/0000-0002-9899-2428>

Final Decision Letter:

Message 24th Aug 2023

:
Dear Dr. Leschziner,

We are now happy to accept your revised paper "Structure of LRRK1 and mechanisms of autoinhibition and activation" for publication as an Article in Nature Structural & Molecular Biology.

Over the next few weeks, your paper will be copyedited to ensure that it conforms to

Nature Structural & Molecular Biology style. Once your paper is typeset, you will receive an email with a link to choose the appropriate publishing options for your paper and our Author Services team will be in touch regarding any additional information that may be required.

As soon as your article is published, you can generate your shareable link by entering the DOI of your article here: http://authors.springernature.com/share. Corresponding authors will also receive an automated email with the shareable link

Your paper will be published online soon after we receive proof corrections and will appear in print in the next available issue. You can find out your date of online publication by contacting the production team shortly after sending your proof corrections. Content is published online weekly on Mondays and Thursdays, and the embargo is set at 16:00 London time (GMT)/11:00 am US Eastern time (EST) on the day of publication. Now is the time to inform your Public Relations or Press Office about your paper, as they might be interested in promoting its publication. This will allow them time to prepare an accurate and satisfactory press release. Include your manuscript tracking number (NSMB-A47389B) and our journal name, which they will need when they contact our press office.

About one week before your paper is published online, we shall be distributing a press release to news organizations worldwide, which may very well include details of your work. We are happy for your institution or funding agency to prepare its own press release, but it must mention the embargo date and Nature Structural & Molecular Biology. If you or your Press Office have any enquiries in the meantime, please contact press@nature.com.

Please note that *Nature Structural & Molecular Biology* is a Transformative Journal (TJ). Authors may publish their research with us through the traditional subscription access route or make their paper immediately open access through payment of an article-processing charge (APC). Authors will not be required to make a final decision about access to their article until it has been accepted. <https://www.springernature.com/gp/open-research/transformative-journals> Find out more about Transformative Journals

Sincerely,

Katarzyna Ciazynska
(she/her)
Associate Editor
Nature Structural & Molecular Biology
<https://orcid.org/0000-0002-9899-2428>
